# Deno-IF: Unsupervised Noisy Visible and Infrared Image Fusion Method

**Han Xu**[1], **Yuyang Li**[1], **Yunfei Deng**[1], **Jiayi Ma**[2], **Guangcan Liu**[1*]

[1] School of Automation, Southeast University, Nanjing, China
[2] Electronic Information School, Wuhan University, Wuhan, China
`{xu_han,yuyangli}@seu.edu.cn`, `{yunfeideng455,jyma2010}@gmail.com`,
`guangcanliu@seu.edu.cn`

## Abstract

Most image fusion methods are designed for ideal scenarios and struggle to handle noise. Existing noise-aware fusion methods are supervised and heavily rely on constructed paired data, limiting performance and generalization. This paper proposes a novel unsupervised noisy visible and infrared image fusion method, comprising two key modules. First, when only noisy source images are available, a convolutional low-rank optimization module decomposes clean components based on convolutional low-rank priors, guiding subsequent optimization. The unsupervised approach eliminates data dependency and enhances generalization across various and variable noise. Second, a unified network jointly realizes denoising and fusion. It consists of both intra-modal recovery and inter-modal recovery and fusion, also with a convolutional low-rankness loss for regularization. By exploiting the commonalities of denoising and fusion, the joint framework significantly reduces network complexity while expanding functionality. Extensive experiments validate the effectiveness and generalization of the proposed method for image fusion under various and variable noise conditions. The code is publicly available at `https://github.com/hanna-xu/Deno-IF`.

## 1 Introduction

The widely used visible images, although rich in color and textures, are vulnerable to environmental factors, *e.g,*, illumination variations and occlusions. In contrast, infrared images can capture thermal radiation information under extreme-light or obscured conditions while they often suffer from poor details and low quality. Therefore, visible and infrared image fusion aims to combine the complementary and valuable information of these two modalities. The single fused image can retain the clarity and fine details of the visible image while incorporating the prominence of thermal targets in the infrared image for comprehensive representation. The fusion results can aid in both the human visual perception and subsequent widespread applications, such as object detection [26, 51, 7], semantic segmentation [20, 43], autonomous driving [47, 3].

With the rapid development of deep learning, many learning-based fusion methods have been proposed. These methods can be categorized into: methods based on auto-encoder (AE) [2, 40], convolutional neural network (CNN) [31, 21, 32], generative adversarial networks (GANs) [11, 42, 18], Transformer [16, 13], Mamba [29] and Diffusion [50, 35, 23]. Despite the progress in network architectures, these methods purely aim to preserve the original information in source images without distinguishing the information quality. In real-world scenarios, the quality of imaging is adversely affected by challenging conditions (*e.g.*, poor illumination) and the inherent constraints

---

[*]Corresponding author

39th Conference on Neural Information Processing Systems (NeurIPS 2025).

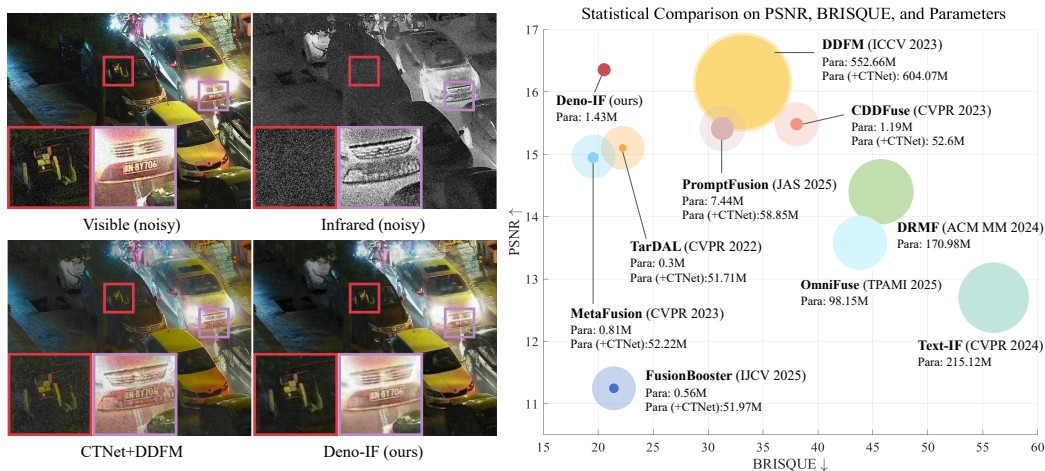

Figure 1: Comparison of Deno-IF with SOTA image denoising and fusion methods by: i) similarity with clean source images (measured by PSNR), ii) image quality assessment (measured by BRISQUE), and iii) complexity (measured by parameter numbers and represented by circle sizes). The high- and low-saturation circle regions represent parameters of fusion and denoising method [25], respectively.

of cost-effective multi-modal devices. These systems are prone to significant noise, stemming from degraded signal acquisition (especially in low-cost microbolometers and CMOS sensors) and ineffective embedded processing pipelines. Under interference conditions, there remain several critical challenges that need to be addressed to achieve robust high-quality fusion results [37, 33].

As most existing fusion methods are tailored for standard scenarios and ineffective at suppressing noise, an intuitive approach is to first denoise the source images before performing fusion. However, the framework taking denoising and fusion as independent tasks presents two limitations. On the one hand, denoising methods are typically deployed on a single source image, making it challenging to leverage complementary knowledge from the other modality. When denoising performance is limited, fusion methods are unable to handle the residual noise or over-smoothness, leading to its persistence in the fusion result, as shown in the result in Fig. 1. On the other hand, the mutually independent framework suffers from significant redundancy. The goal of image fusion is to extract vital and complementary information, which is then integrated into the fused images. Besides, image denoising aims to mine essential clean data from noisy inputs. From the perspective of essential information extraction and mining, their objectives are consistently aligned to some extent. The independent framework ignores the commonality, leading to redundancy as Fig. 1. There remains the potential to jointly realize denoising and fusion in a streamlined manner.

Although some degradation-aware image fusion methods exist [22, 36], they are supervised methods aimed at learning the mapping from degraded source images to clean fused images. The key lies in the construction of large amount of noisy and clean source image pairs. Their denoising effectiveness is limited by the fixed and limited mapping relationship between the constructed noisy and clean data, resulting in poor generalization. As a result, these methods are only effective for specific types of noise with particular variances. Facing various and variable noise encountered in real-world scenarios, these methods are highly susceptible to failure, as shown in their performances in Fig. 1.

To overcome these issues, this paper proposes an unsupervised noisy visible and infrared image fusion method, termed as Deno-IF. It consists of two modules, including a convolutional low-rank optimization module and a joint denoising and fusion module. First, based on the convolutional low-rank property of high-quality data, the convolutional low-rank optimization module decomposes clean component from noisy input through convolution nuclear norm minimization in an unsupervised manner. Then, the joint denoising and fusion module takes noisy source images as input and directly outputs the fused image. The network includes both intra-modal recovery and inter-modal recovery and fusion, with self- and cross-modal attention to deal with complex individual-modal and complementary multi-modal information. The decomposed data provides the optimization guidance for the joint denoising and fusion module. Besides, a convolution matrix-based regularization loss is

designed to further suppress noise in fused images. Thus, it can realize denoising and fusion jointly and generate clean fused images. The contributions are summarized as:

• We propose an unsupervised noisy visible and infrared image fusion method. Without the supervision of clean data, it can still realize denoising and fusion simultaneously with fewer network parameters, and is robust against various and variable noise conditions, ensuring effective across diverse scenarios.

• For noisy source images, we design a convolutional low-rank optimization module. As clean data usually exhibits convolutional low-rankness, we introduce the convolution nuclear norm minimization to decompose clean data from noisy inputs. It provides the optimization guidance for fusion network in the training phase.

• For joint denoising and fusion, a Transformer-based network consisting of intra-modal recovery and inter-modal recovery and fusion is designed. It leverages self- and cross-modal attention to collaboratively strive to approximate the optimization guidance. Besides, a convolution matrix-based regularization loss is designed to further suppress noise in fused images.

## 2    Related Work

**Learning-Based Image Fusion Methods**. These methods preserve the information in source images through the design of network architectures and loss functions. According to architectures, they can be divided into AE, CNN, GAN, Transformer, Mamba and Diffusion-based methods. AE-based methods design encoders and decoders for feature extraction and reconstruction, with traditional or learning-based [30] fusion strategies. CNN-based methods utilize end-to-end fusion [41, 6] and the key lies in pixel-level loss functions. GAN-based methods preserve unique information through adversarial games from the perspective of probability distribution while they always suffer from unstable training and model collapse [12, 17]. Transformer and Mamba-based methods build long-range dependency to solve the limited receptive field of CNN [13, 24, 29, 15]. Diffusion-based methods leverages the powerful generative properties of diffusion models to establish a stable fusion process [38, 4, 34]. However, these methods are designed for normal scenarios and ignore degradations in real world. Facing degraded images, they fail to tackle degradations, reducing the quality and usability of results.

**Degradation-Aware Image Fusion Methods**. Some fusion methods suppress degradations and achieve fusion in a unified network. Text-IF [36] leverages the text prompts with degradation information to guide the enhanced fusion. The network is trained with the manually constructed paired degraded and high-quality source images. DRMF [22] pretrains multiple degradation robust diffusion models and devises a combination module to integrate priors. Similarly, the performance also depends on paired constructed data. They can handle different types of degradations including noise. However, since the introduced noise has fixed variance, it demonstrates limited generalization. These methods heavily rely on paired constructed data. This paper proposes an unsupervised method that requires only degraded data for training. We infer the clean component to guide the fusion network, significantly reducing the data demand and the number of network parameters.

## 3    Proposed Method

### 3.1    Problem Formulation

Mathematically, a pair of observed noisy visible and infrared images $\{\mathbf{V}, \mathbf{R}\}$ can be decomposed as:

$$\mathbf{V} = \mathbf{L}_v + \mathbf{S}_v, \quad \mathbf{R} = \mathbf{L}_r + \mathbf{S}_r, \tag{1}$$

where $\mathbf{L}_v, \mathbf{L}_r$ is the clean data and $\mathbf{S}_v, \mathbf{S}_r$ is the noise. As clean data, $\mathbf{L}_v, \mathbf{L}_r$ should be of convolutional low-rankness [8]. $\mathbf{S}_v, \mathbf{S}_r$ should be small-scale perturbations with limited overall energy and amplitude to ensure the convolutional low-rankness of $\mathbf{L}_v, \mathbf{L}_r$.

Taking noisy source images $\mathbf{V}, \mathbf{R}$ as input, the jointly denoising and fusion module generates a high-quality fused image $\mathbf{F}$ free from noise. Based on the decomposition assumption in Eq. (1), the goal is equivalent to making $\mathbf{F}$ contain vital information in $\mathbf{L}_v, \mathbf{L}_r$ mined from $\mathbf{V}, \mathbf{R}$, respectively.

As shown in Fig. 2, the overall framework consists of the convolutional low-rank optimization module and the joint denoising and fusion module. The convolutional low-rank optimization module solves

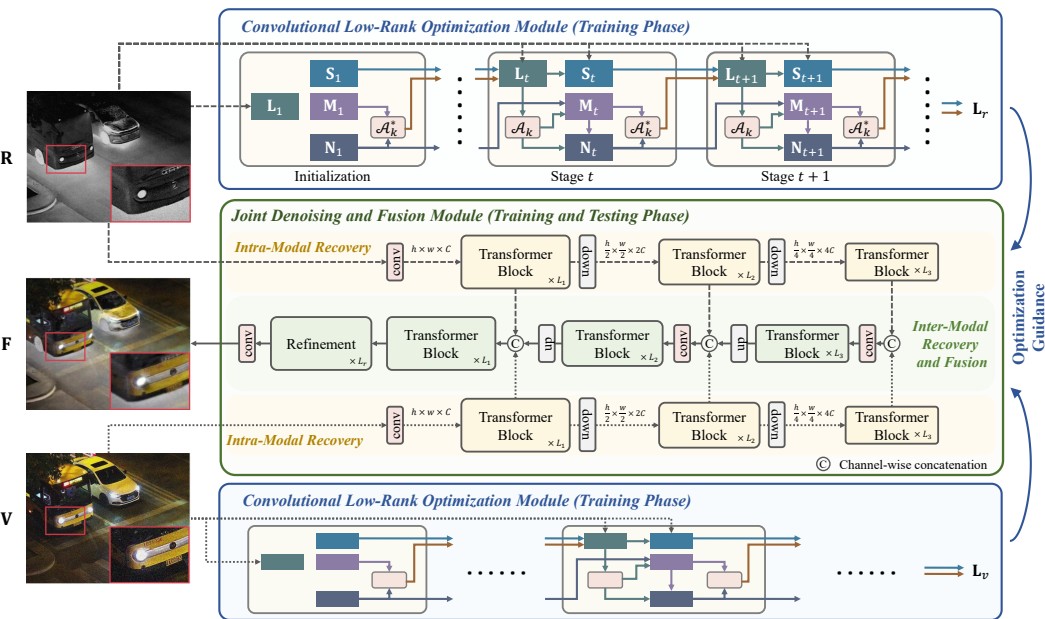

Figure 2: Overall framework of the unsupervised noisy infrared and visible image fusion method.

the ill-posed decomposition problem to provide precise guidance for network optimization. In the joint denoising and fusion module, a network $\mathcal{F}$ takes noisy source image as input, consists of intra-modal recovery and inter-modal recovery and fusion. Based on Transformer, intra-modal recovery leverages the self-attention mechanism to enhance restoration by capturing long-range dependencies within individual modality. Inter-modal recovery and fusion exploits cross-modal attention, enabling the joint restoration and fusion of complementary multi-modal information. $\mathcal{F}$ is optimized with the results of the convolutional low-rank optimization module to realize joint denoising and fusion.

## 3.2 Convolutional Low-Rank Optimization Module

For the decomposition problem in Eq. (1), we take the infrared image $\mathbf{R} \in \mathbb{R}^{h \times w}$ as example (subscript $r$ is temporarily omitted for simplification). To ensure the respective properties of $\mathbf{L}$ and $\mathbf{S}$, the problem can be formulated as:

$$\min_{\mathbf{L},\mathbf{S}} \|\mathcal{A}_k(\mathbf{L})\|_* + \beta\|\mathbf{S}\|_F^2, \quad s.t. \ \mathbf{R} = \mathbf{L} + \mathbf{S}, \tag{2}$$

where the first term is based on convolution nuclear norm, which is the nuclear norm of its convolution matrix [8]. $\mathcal{A}_k(\cdot)$ is a linear map from $\mathbb{R}^{h \times w}$ to $\mathbb{R}^{hw \times k_1 k_2}$ where $k_1$ ($1 \leq k_1 \leq h$) and $k_2$ ($1 \leq k_2 \leq h$) denote the kernel size in $\mathcal{A}_k(\cdot)$. It produces the convolution matrix of the input. $\|\cdot\|_*$ and $\|\cdot\|_F$ denote nuclear and Frobenius norms, respectively. $\beta$ controls the trade-off between these terms.

Within the maximum a posteriori (MAP) inference framework, $\mathbf{L}$ and $\mathbf{S}$ can be obtained by minimizing the following energy function:

$$E(\mathbf{L},\mathbf{S}) = \|\mathbf{R} - \mathbf{L} - \mathbf{S}\|_F^2 + \alpha\|\mathcal{A}_k(\mathbf{L})\|_* + \beta\|\mathbf{S}\|_F^2, \tag{3}$$

where the first term is fidelity term and the last two terms are regularization term denoting imposed priors over $\mathbf{L}$ and $\mathbf{S}$. $\alpha$ also controls the trade-off.

With [19] releasing nuclear norm as:

$$\|\mathbf{X}\|_* = \min_{\mathbf{A},\mathbf{B}} \frac{1}{2}\|\mathbf{A}\|_F^2 + \frac{1}{2}\|\mathbf{B}\|_F^2, \quad s.t. \ \mathbf{X} = \mathbf{AB}, \tag{4}$$

the energy function can be rewritten as:

$$E(\mathbf{L},\mathbf{S},\mathbf{M},\mathbf{N}) = \|\mathbf{R} - \mathbf{L} - \mathbf{S}\|_F^2 + \frac{\alpha}{2}\|\mathbf{M}\|_F^2 + \frac{\alpha}{2}\|\mathbf{N}\|_F^2 + \gamma\|\mathcal{A}_k(\mathbf{L}) - \mathbf{MN}\|_F^2 + \beta\|\mathbf{S}\|_F^2, \tag{5}$$

where $\mathbf{M} \in \mathbb{R}^{hw \times m}, \mathbf{N} \in \mathbb{R}^{m \times k_1 k_2}$. $\gamma$ is a trade-off parameter.

This problem can be solved by iteratively addressing the subproblems related to $\mathbf{L}, \mathbf{S}, \mathbf{M}, \mathbf{N}$. With $t$ denotes the iteration step, the problem can be partitioned into the following four subproblems:

$$\mathbf{L}_t = \arg\min_{\mathbf{L}} \|\mathbf{R} - \mathbf{L} - \mathbf{S}_{t-1}\|_F^2 + \gamma \|\mathcal{A}_k(\mathbf{L}) - \mathbf{M}_{t-1}\mathbf{N}_{t-1}\|_F^2, \tag{6a}$$

$$\mathbf{S}_t = \arg\min_{\mathbf{S}} \|\mathbf{R} - \mathbf{L}_t - \mathbf{S}\|_F^2 + \beta \|\mathbf{S}\|_F^2, \tag{6b}$$

$$\mathbf{M}_t = \arg\min_{\mathbf{M}} \frac{\alpha}{2} \|\mathbf{M}\|_F^2 + \gamma \|\mathcal{A}_k(\mathbf{L}_t) - \mathbf{M}\mathbf{N}_{t-1}\|_F^2, \tag{6c}$$

$$\mathbf{N}_t = \arg\min_{\mathbf{N}} \frac{\alpha}{2} \|\mathbf{N}\|_F^2 + \gamma \|\mathcal{A}_k(\mathbf{L}_t) - \mathbf{M}_t\mathbf{N}\|_F^2. \tag{6d}$$

When t=1, we initially set the variables as $\mathbf{L}_1 = \mathbf{R}$. $\mathbf{S}_1$ is set to randomly distributed noise. $\mathbf{M}_1$ and $\mathbf{N}_1$ are set to identity-like diagonal matrix. During the iteration process, the closed-form solutions for these four subproblems can then be obtained as:

$$\mathbf{L}_t = \frac{\mathbf{R} - \mathbf{S}_{t-1} + \gamma \mathcal{A}_k^*(\mathbf{M}_{t-1}\mathbf{N}_{t-1})}{\gamma + 1}, \tag{7}$$

where $\mathcal{A}_k^*(\cdot)$ is the Hermitian adjoint of $\mathcal{A}_k$.

$$\mathbf{S}_t = \frac{\mathbf{R} - \mathbf{L}_t}{\beta + 1}, \quad \mathbf{M}_t = 2\gamma \mathcal{A}_k(\mathbf{L}_t)\mathbf{N}_{t-1}^\top(\alpha\mathbf{I} + 2\gamma\mathbf{N}_{t-1}\mathbf{N}_{t-1}^\top)^{-1}, \tag{8}$$

where $\mathbf{I}$ is a $m \times m$ identity matrix.

$$\mathbf{N}_t = 2\gamma(\alpha\mathbf{I} + 2\gamma\mathbf{M}_t^\top\mathbf{M}_t)^{-1}\mathbf{M}_t^\top \mathcal{A}_k(\mathbf{L}_t). \tag{9}$$

For the visible image $\mathbf{V} \in \mathbb{R}^{h \times w \times c}$, the primary distinction to infrared data lies in three channels. It results in slight differences in the construction of convolution matrix and the shapes of $\mathbf{M}, \mathbf{N}$. Specifically, $\mathcal{A}_k(\mathbf{V})$ is a linear map from $\mathbb{R}^{h \times w \times c}$ to $\mathbb{R}^{hwc \times k_1 k_2 k_3}$, where $k_1$ $(1 \leq k_1 \leq h)$, $k_2$ $(1 \leq k_2 \leq w)$ and $k_3$ $(1 \leq k_3 \leq c)$ are the kernel size. $\mathbf{M} \in \mathbb{R}^{hwc \times n}$ and $\mathbf{N} \in \mathbb{R}^{n \times k_1 k_2 k_3}$.

### 3.3 Joint Denoising and Fusion Module

**Network Architecture**. Directly denoising multi-modal source images as a whole without distinction will overlook the modal differences and noise characteristics between multi-modal images, posing greater challenges to $\mathcal{F}$. Thus, the fusion network $\mathcal{F}$ comprises two key components: intra-modal recovery and inter-modal recovery and fusion. In one source image, the modality is unique and the noise characteristics are consistent. We first perform intra-modal feature recovery in visible and infrared domains, respectively. Then, the inter-modal recovery and fusion integrates information across different modalities while effectively handling the complex noise characteristics among these modalities. The overall network is named as I2Former. The output is the clean fused image $\mathbf{F}$.

For the two components in $\mathcal{F}$, we adopt the Transformer block [39] for intra- and inter-modal feature extraction and integration. In intra-modal recovery, we respectively extract features from $\mathbf{V}, \mathbf{R}$ as $\mathbf{f}_v, \mathbf{f}_r$. In each block, the intra-modal multi-Dconv head transposed attention ($T_{ter}$) generates query, key and value projections for $\mathbf{f}_v, \mathbf{f}_r$ as:

$$\{\mathbf{Q}_v, \mathbf{K}_v, \mathbf{V}_v\} = T_{ter}^V(\mathbf{f}_v), \quad \{\mathbf{Q}_r, \mathbf{K}_r, \mathbf{V}_r\} = T_{ter}^R(\mathbf{f}_r). \tag{10}$$

Then, the output visible feature maps are $\hat{\mathbf{f}}_v = \text{Softmax}(\frac{\mathbf{K}_v \cdot \mathbf{Q}_v}{\varsigma})\mathbf{V}_v + \mathbf{f}_v$, where $\varsigma$ is a learnable scaling parameter. Infrared features are modulated similarly. The features are passed through gated-Dconv feed-forward network for controllable transformation. Multiple blocks generate multi-scale features.

In inter-modal recovery and fusion, the visible and infrared features of the smallest scale are concatenated, fed into a convolutional layer for channel reduction and upsampled. With $\mathbf{f}_f$ and $\mathbf{f}_v, \mathbf{f}_r$ as input, the inter-modal multi-Dconv head transposed attention ($T_{tra}$) generates projections as:

$$\left\{\mathbf{Q}_f, \mathbf{K}_f, \mathbf{V}_f\right\} = T_{tra}\left(W\left(\mathbf{f}_v, \mathbf{f}_r, \mathbf{f}_f\right)\right), \tag{11}$$

where $W$ is a convolutional layer for channel reduction. The final output features can be obtained as $\hat{\mathbf{f}}_f = \text{Softmax}(\frac{\mathbf{K}_f \cdot \mathbf{Q}_f}{\varsigma})\mathbf{V}_f + W(\mathbf{f}_v, \mathbf{f}_r, \mathbf{f}_f)$. The fused features of the original scale are finally passed through some blocks for refinement and a convolutional layer to generate the fused image $\mathbf{F}$.

**Loss Functions**. From the aspect of network optimization, clean components derived with convolutional low-rank priors serve as i) physics-driven prior injector to regularize fused images, and ii) implicit teachers that enable knowledge distillation of denoising priors into the fusion network. It enables simultaneous noise suppression and information fusion. Thus, the loss functions of $\mathcal{F}$ consist of some data fidelity terms and a regularization term.

*Data Fidelity Terms*. Sec. 3.2 obtains the estimation of $\mathbf{L}_v$ and $\mathbf{L}_r$. The data fidelity terms constrain the fused image $\mathbf{F}$ to keep similarity with $\mathbf{L}_v$ and $\mathbf{L}_r$ from multiple perspectives to preserve the scene information, including intensity, gradients, and chrominance components.

The intensity loss makes $\mathbf{F}$ capture proper intensity distribution. One target is to preserve the thermal prominence in $\mathbf{L}_r$. Considering the low thermal radiation regions in $\mathbf{L}_r$ and low-light regions in $\mathbf{L}_v$, the other target is to preserve the more prominent information. The intensity loss is denoted as $\mathcal{L}_{in} = \|\mathbf{F}^y - \max(\mathbf{L}_v^y, \mathbf{L}_r)\|_1$ where $\mathbf{L}_v^y, \mathbf{F}^y$ are the Y channels of $\mathbf{L}_v, \mathbf{F}$ in YCbCr space, respectively.

The gradient loss preserves the prominent textures in $\mathbf{L}_v, \mathbf{L}_r$ to present more contents. We generate a gradient mask $M_g$ considering the gradient relationship between $\mathbf{L}_v^y, \mathbf{L}_r$. When $\left|\nabla \mathbf{L}_{v i,j}^y\right| > \left|\nabla \mathbf{L}_{r i,j}\right|$, $\mathbf{m}_{g i,j} = 1$ where $\nabla$ is Laplacian operator, $i$ and $j$ indicate the spatial position. Otherwise, $\mathbf{m}_{g i,j} = 0$. The loss constrains the gradient similarity as:

$$\mathcal{L}_g = \|\nabla \mathbf{F}^y - [\mathbf{m}_g \cdot \nabla \mathbf{L}_v^y + (1 - \mathbf{m}_g) \cdot \nabla \mathbf{L}_r]\|_1. \tag{12}$$

The chrominance loss preserves the chrominance information in $\mathbf{L}_v$ into $\mathbf{F}$. $\mathbf{L}_v, \mathbf{F}$ are translated into YCbCr space and the constraint is performed on chrominance channels as:

$$\mathcal{L}_{chr} = \|\mathbf{F}^{Cb} - \mathbf{L}_v^{Cb}\|_1 + \|\mathbf{F}^{Cr} - \mathbf{L}_v^{Cr}\|_1. \tag{13}$$

*Regularization Term*. In the optimization process of $\mathbf{F}$, the performance gap in intra-modal recovery and the residual and cumulative deviation in inter-modal recovery and fusion will affect performance. To further correct deviations, the convolutional low rank of fused image is regularized as:

$$\mathcal{L}_{rank} = \|\mathcal{A}_k(\mathbf{F})\|_*, \tag{14}$$

where $\mathcal{A}_k(\mathbf{F})$ computes the convolution matrix of $\mathbf{F}$. $\mathcal{A}_k(\cdot)$ is a linear map from $\mathbb{R}^{h \times w \times c}$ to $\mathbb{R}^{hwc \times k_1 k_2 k_3}$ (defined at the end of Sec. 3.2). The nuclear norm imposes a low-rank constraint on the convolution matrix of $\mathbf{F}$, thereby reducing residual noise in $\mathbf{F}$ by mining its intrinsic structure.

With hyper-parameters $\lambda, \eta, \kappa$ to control the trade-off, the final loss function is defined as:

$$\mathcal{L}_{\mathcal{F}} = \mathcal{L}_{in} + \lambda \mathcal{L}_g + \eta \mathcal{L}_{chr} + \kappa \mathcal{L}_{rank}. \tag{15}$$

## 4 Experiments and Results

**Datasets and Implementation.** Existing datasets are collected for the target of high image quality, thus relying on cost-prohibitive imaging systems to ensure high signal-to-noise ratios. As there are no publicly available noisy infrared and visible datasets for direct use, We simulate real-life noisy source images by injecting two representative types of noise with various levels (additive Gaussian noise and multiplicative speckle noise) into clean data [27, 46].The clean data is not used during training. We train Deno-IF on 2120 pairs of visible and infrared images across two datasets, including LLVIP [5] and M3FD [9]. For the introduced noise, $\sigma$ randomly distributed in $[10, 50]$. In the training phase, images are randomly cropped into patches of $128 \times 128$ for training. The evaluation is performed on image pairs introduced Gaussian or speckle noise with randomly distributed variance.

In convolutional low-rank optimization module, $h$ and $w$ are 128, $c = 3$. Kernel size of $\mathcal{A}_k(\cdot)$ are $k_1, k_2 = 12, k_3 = 2$. $m, n = 256$. We perform 30 iterations for each patch. For parameters in Eq. (5), $\beta = 2$. $\alpha, \gamma$ are empirically related to data characteristics. As an unsupervised method, for a patch $\mathbf{x}$, we roughly estimate its noise level as $n = \mathbb{E}\left[|\nabla(\mathbf{x}) - \nabla(G(\mathbf{x}))|\right]$, where $G(\cdot)$ is Gaussian blur. $\alpha$ is $200n$ and $80n$ for visible and infrared data respectively. $\beta = 15n$. These manually designed settings stem from the consideration of i) a principled trade-off for adaptive regularization (strong for high-level noise for noise suppression, weak for low-level noise to preserve details); and ii) computational efficiency for an optimal efficiency-accuracy balance.

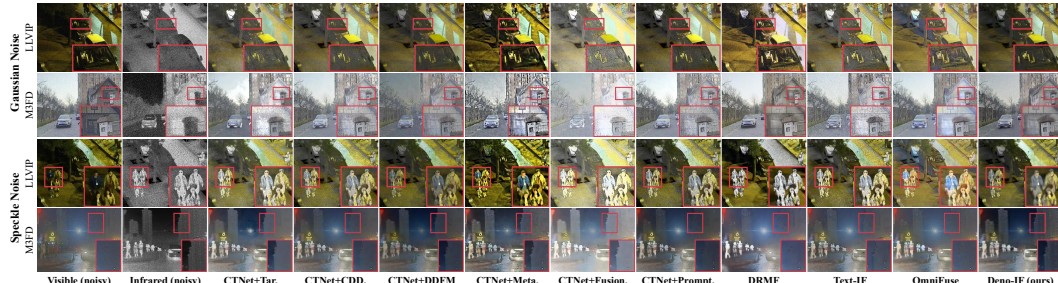

Figure 3: Qualitative results when both source images are subject to various-level **Gaussian/speckle** noise on LLVIP and M3FD datasets.

Table 1: Quantitative results when both source images suffer from various-level **Gaussian** or **Speckle** noise (**bold**: optimal, underline: suboptimal).

| Gaussian | LLVIP Dataset | | | | | M3FD Dataset | | | | |
|---|---|---|---|---|---|---|---|---|---|---|
| Metrics | SSIM↑ | PSNR↑ | FSIM↑ | CC↑ | BRISQUE↓ | SSIM↑ | PSNR↑ | FSIM↑ | CC↑ | BRISQUE↓ |
| CT.+Tar. | 0.421±0.169 | 15.600±1.464 | 0.792±0.024 | 0.680±0.082 | 19.519±8.577 | 0.469±0.143 | 14.597±2.781 | 0.756±0.032 | 0.513±0.195 | 24.898±9.713 |
| CT.+CDD. | 0.446±0.206 | 15.881±1.366 | 0.803±0.028 | 0.698±0.072 | 33.929±20.019 | 0.466±0.140 | 15.076±2.789 | 0.772±0.031 | 0.501±0.226 | 42.196±14.175 |
| CT.+DDFM | 0.474±0.178 | 16.782±1.138 | **0.815±0.022** | 0.736±0.077 | 30.624±16.596 | 0.489±0.165 | 15.525±2.513 | 0.767±0.051 | **0.587±0.191** | 35.771±13.177 |
| CT.+Meta. | 0.502±0.099 | 15.274±1.193 | 0.694±0.045 | 0.662±0.083 | 24.597±12.281 | 0.533±0.132 | 14.606±2.650 | 0.675±0.053 | 0.548±0.194 | **14.430±5.869** |
| CT.+Fusion. | 0.270±0.135 | 11.694±0.624 | 0.760±0.042 | 0.679±0.078 | 26.381±11.446 | 0.459±0.094 | 10.618±1.040 | 0.744±0.047 | 0.565±0.158 | 16.474±11.822 |
| CT.+Prompt. | 0.433±0.197 | 15.811±1.331 | 0.803±0.029 | 0.702±0.073 | 36.876±19.142 | 0.474±0.125 | 15.007±2.679 | 0.777±0.033 | 0.470±0.243 | 25.837±13.808 |
| DRMF | 0.148±0.081 | 11.899±1.242 | 0.693±0.031 | 0.489±0.156 | 67.922±12.462 | 0.280±0.121 | 13.482±1.628 | 0.738±0.043 | 0.334±0.219 | 44.186±11.117 |
| Text-IF | 0.208±0.069 | 14.956±1.125 | 0.770±0.025 | 0.652±0.071 | 54.482±10.614 | 0.225±0.089 | 13.837±2.509 | 0.710±0.055 | 0.404±0.209 | 37.228±8.241 |
| OmniFuse | 0.377±0.080 | 12.642±0.727 | 0.731±0.026 | 0.642±0.089 | 41.046±3.748 | 0.540±0.098 | 14.488±2.119 | 0.763±0.031 | 0.459±0.211 | 46.787±6.876 |
| Deno-IF | **0.616±0.064** | **17.070±1.426** | 0.811±0.023 | **0.738±0.074** | **16.725±4.091** | **0.585±0.081** | **15.637±2.899** | **0.788±0.029** | 0.563±0.205 | 24.298±7.095 |

| Speckle | LLVIP Dataset | | | | | M3FD Dataset | | | | |
|---|---|---|---|---|---|---|---|---|---|---|
| Metrics | SSIM↑ | PSNR↑ | FSIM↑ | CC↑ | BRISQUE↓ | SSIM↑ | PSNR↑ | FSIM↑ | CC↑ | BRISQUE↓ |
| CT.+Tar. | 0.423±0.073 | 15.125±1.464 | 0.776±0.024 | 0.687±0.072 | 13.375±9.263 | 0.553±0.129 | 15.358±2.393 | 0.775±0.039 | 0.553±0.206 | 13.950±9.283 |
| CT.+CDD. | 0.419±0.096 | 15.527±1.515 | 0.789±0.026 | 0.681±0.063 | 45.497±13.496 | 0.539±0.145 | 15.906±2.520 | 0.789±0.041 | 0.558±0.215 | 25.231±14.022 |
| CT.+DDFM | 0.474±0.086 | 16.191±1.218 | **0.808±0.022** | 0.723±0.069 | 32.897±12.572 | 0.584±0.125 | 16.232±2.259 | 0.806±0.047 | 0.641±0.157 | 18.320±11.767 |
| CT.+Meta. | 0.538±0.072 | 15.102±1.220 | 0.697±0.043 | 0.669±0.075 | 10.860±9.531 | 0.615±0.115 | 15.606±2.526 | 0.710±0.046 | 0.611±0.164 | **10.412±6.587** |
| CT.+Fusion. | 0.338±0.055 | 11.909±0.694 | 0.737±0.032 | 0.683±0.066 | 29.446±7.250 | 0.516±0.106 | 10.950±0.807 | 0.774±0.034 | 0.612±0.150 | 21.178±9.460 |
| CT.+Prompt. | 0.381±0.085 | 15.212±1.313 | 0.787±0.026 | 0.681±0.066 | 48.286±12.442 | 0.522±0.146 | 15.881±2.447 | 0.784±0.041 | 0.521±0.244 | 21.837±12.825 |
| DRMF | 0.475±0.047 | 12.417±1.377 | 0.722±0.021 | 0.567±0.137 | 21.406±10.566 | 0.649±0.100 | 14.147±2.561 | 0.790±0.026 | 0.423±0.209 | 43.180±17.155 |
| Text-IF | 0.400±0.054 | 13.953±1.119 | 0.759±0.019 | 0.665±0.060 | 27.072±9.667 | 0.554±0.128 | 15.941±2.402 | 0.780±0.037 | 0.496±0.235 | 18.908±10.381 |
| OmniFuse | 0.479±0.052 | 12.859±0.775 | 0.741±0.017 | 0.665±0.074 | 39.696±4.430 | 0.585±0.104 | 15.221±2.089 | 0.767±0.033 | 0.523±0.213 | 44.850±7.131 |
| Deno-IF | **0.633±0.052** | **16.528±1.370** | 0.805±0.022 | **0.738±0.065** | **9.280±5.918** | **0.656±0.075** | **16.525±2.800** | **0.813±0.026** | **0.648±0.155** | 16.514±8.460 |

In the joint denoising and fusion module, I2Former is updated with Adam Optimizer with batch size as 4 and epoch as 80. Learning rate is 2e-4 with exponential decay. Hyper-parameters are $\lambda = 1e3$, $\eta = 30$. $\kappa$ increases during training and equals the multiplication of 1e-6 and epoch. Numbers of blocks in Fig. 2 are $L_r, L_3 = 2, L_1, L_2 = 4$. Experiments are conducted on an NVIDIA 3090 GPU.

**Comparison Results.** We consider two typical types of noise (additive and multiplicative noise) during imaging process, *e.g.*, Gaussian and speckle noise in source images. For noisy source images, we compare Deno-IF with both the combination of SOTA denoising and fusion methods and degradation-aware image fusion methods. As some fusion methods (TarDAL [9], CDDFuse [49], DDFM [50], MetaFusion [48], FusionBooster [1], and PromptFusion [10]) cannot handle noise, CTNet [25] (supervised denoising method) is used for pre-denoising before these fusion methods are performed. The rationale of selecting CTNet for comparison is two-fold: i) validating the unsupervised generalization of the proposed method against the supervised denoising performance of CTNet; ii) validating the improvements of architecturally relevant frameworks and objectives as CTNet also designs a Transformer-based network. The supervised degradation-aware fusion methods (DRMF [22], Text-IF [36], and OmniFuse [44]) directly fuse noisy source images.

*Qualitative Results.* Qualitative results where both source images suffer from various-level Gaussian or speckle noise across two datasets are shown in Fig. 3. First, as Deno-IF jointly realizes denoising and fusion, it avoids the residual noise in competitors caused by the limited pre-denoising performance. Second, as an unsupervised method, Deno-IF can infer clean data from source images which both suffer from noise and present clear fused images. On the other hand, it shows robust

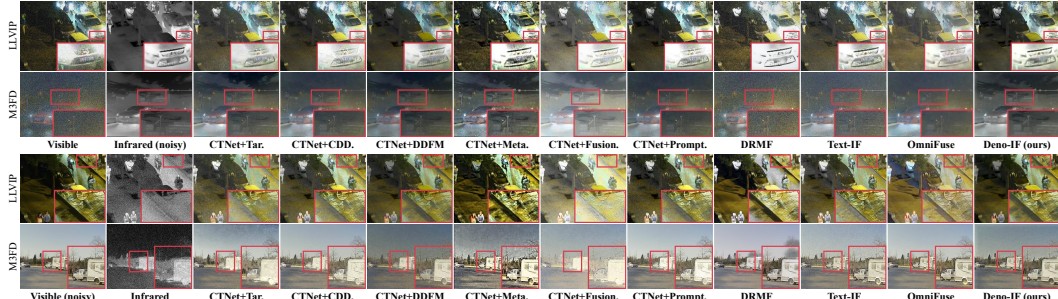

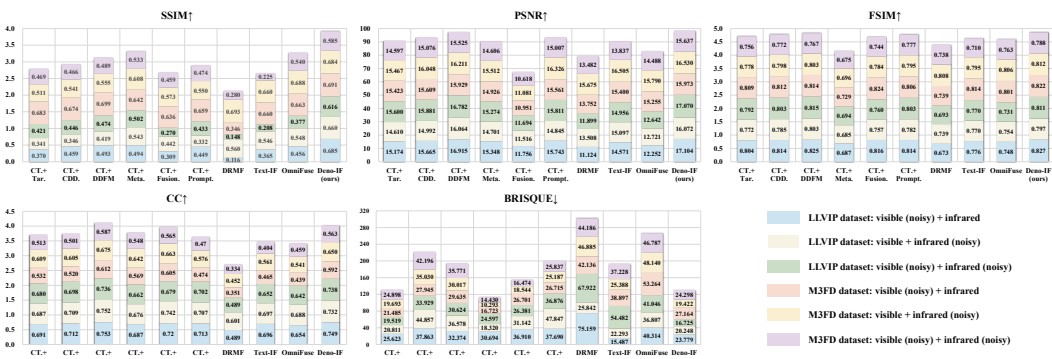

Figure 4: Qualitative results when only one source image is subject to various-level noise.

Figure 5: Quantitative results in difference situations of noise presence on two datasets (see *Supplementary Material* for results in specific situations).

performance across various noise types and levels, addressing the low generalization of existing supervised denoising and fusion methods. Finally, the inference on clean data in Deno-IF is based on convolutional low-rankness, avoiding excessive distortion of meaningful scene content in competitors.

*Quantitative Results.* Traditional fusion metrics fall into similarity-based metrics (evaluating similarity between fused and source images) and statistics-based metrics (measuring image characteristics, e.g., SD, SF, EN, etc.). However, when fusing noisy images, statistics-based metrics may produce misleading results as noise may artificially inflate these values (e.g., higher EN/SF caused by noise rather than meaningful information). Thus, in this work, we prioritize similarity-based full-reference metrics (*i.e.*, SSIM [28], PSNR, feature similarity index (FSIM) [45] and correlation coefficient (CC)) and a quality-based no-reference metric BRISQUE [14] to evaluate denoising and fusion performances on 30 image pairs.

As reported in Tab. 1, Deno-IF achieves optimal performances on most metrics. The results on full-reference metrics indicate that our results show high similarity with clean source images. The results on the non-reference metric indicate that our results have less distortion. Finally, the comprehensive results and small variances indicate the generalization of Deno-IF for various scenes, noise types, and noise levels.

**Validation on Single-Modal Noise.** To validate the generalization under different noise presence situations especially specific-modal noise, we perform experiments where only one source image suffers from noise. We take source images with Gaussian noise for example and qualitative results are shown in Fig. 4. On the one hand, Deno-IF effectively removes noise in infrared or visible modalities without being affected by modal differences, avoiding residual noise in results. On the other hand, it indicates the generalization of Deno-IF in different noise presence situations. As reported in the quantitative results of each situation in Fig. 5, Deno-IF shows the optimal results in SSIM, PSNR and FSIM, and suboptimal results in CC and BRISQUE in total amount. From the perspective of variance across situations and datasets, Deno-IF achieves advantageous and balanced performances.

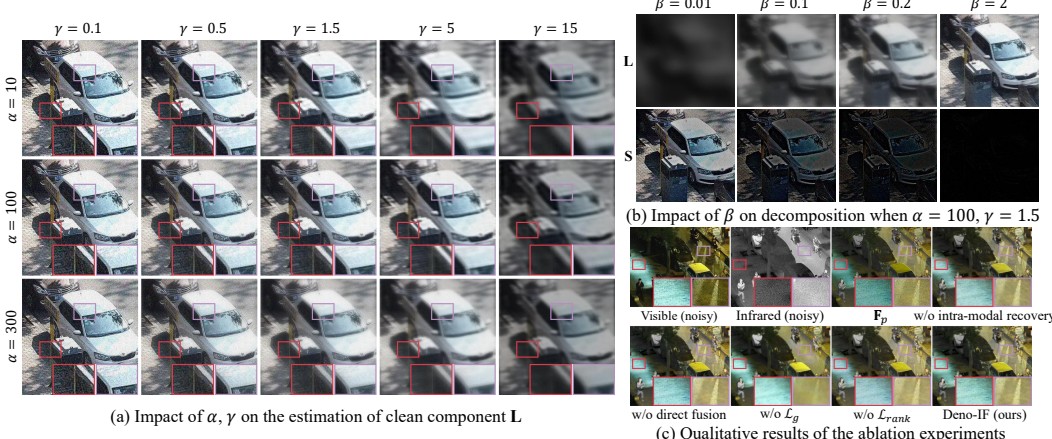

Figure 6: Impacts of decomposition parameter settings and results of ablation experiments.

Table 2: Quantitative results of ablation study and performances on dealing with different-level noise.

| Metrics | SSIM↑ | PSNR↑ | FSIM↑ | CC↑ |
|---|---|---|---|---|
| w/o intra-modal | 0.606 | 16.960 | 0.799 | 0.730 |
| w/o direct fusion | 0.613 | 16.965 | 0.797 | 0.730 |
| w/o $\mathcal{L}_g$ | 0.600 | 16.160 | 0.789 | 0.685 |
| w/o $\mathcal{L}_{rank}$ | 0.611 | 16.554 | 0.802 | 0.727 |
| Deno-IF (ours) | **0.616** | **17.070** | **0.811** | **0.738** |

(a) Results of ablation study.

| | PSNR↑ | | | | SSIM↑ | | | | FSIM↑ | | | | CC↑ | | | | BRISQUE↓ | | | |
|---|---|---|---|---|---|---|---|---|---|---|---|---|---|---|---|---|---|---|---|---|
| $\sigma$ | DRMF | Text. | Omn. | Deno. | DRMF | Text. | Omn. | Deno. | DRMF | Text. | Omn. | Deno. | DRMF | Text. | Omn. | Deno. | DRMF | Text. | Omn. | Deno. |
| 10 | 11.997 | 16.141 | 12.522 | 16.878 | 0.309 | 0.431 | 0.484 | 0.675 | 0.716 | 0.814 | 0.747 | 0.818 | 0.557 | 0.702 | 0.667 | 0.740 | 26.855 | 33.060 | 41.297 | 15.364 |
| 20 | 11.874 | 15.736 | 12.551 | 17.097 | 0.182 | 0.239 | 0.427 | 0.650 | 0.705 | 0.793 | 0.738 | 0.816 | 0.528 | 0.681 | 0.657 | 0.738 | 58.717 | 45.843 | 41.610 | 16.493 |
| 30 | 11.949 | 14.433 | 12.701 | 17.199 | 0.170 | 0.148 | 0.384 | 0.632 | 0.700 | 0.737 | 0.732 | 0.813 | 0.510 | 0.629 | 0.648 | 0.735 | 66.016 | 58.087 | 40.411 | 17.989 |
| 40 | 11.935 | 14.182 | 12.803 | 17.262 | 0.138 | 0.109 | 0.350 | 0.617 | 0.688 | 0.725 | 0.728 | 0.809 | 0.467 | 0.604 | 0.636 | 0.733 | 74.548 | 61.441 | 38.531 | 20.514 |
| 50 | 11.719 | 13.477 | 12.848 | 17.242 | 0.106 | 0.084 | 0.318 | 0.602 | 0.675 | 0.693 | 0.722 | 0.805 | 0.423 | 0.562 | 0.624 | 0.732 | 75.814 | 66.258 | 35.604 | 24.256 |

(b) Results on dealing with different-level noise.

**Optimization Parameter Settings.** We validate impacts of $\alpha, \beta, \gamma$ in Eq. (5) on the estimation of clean component $\mathbf{L}$. We first analysis impacts of $\alpha, \gamma$ and then analyze the impact of $\beta$ with fixed $\alpha, \gamma$. Taking visible data for example, when $\beta = 2$, Fig. 6a shows results with varying $\alpha, \gamma$. $\alpha$ constrains the Frobenius norm of $\mathbf{M}, \mathbf{N}$. $\gamma$ constrains similarity between $\mathcal{A}_k(\mathbf{L})$ and $\mathbf{MN}$. Small $\gamma$ fails to ensure similarity. Regardless of varying $\alpha$, $\mathbf{L}$ tends to be noisy. As $\gamma$ increases, $\mathcal{A}_k(\mathbf{L})$ tends to be low-rank for clarify. Excessively large $\gamma$ disrupts the similarity between $\mathbf{L} + \mathbf{S}$ and noisy image, leading to brightness and content distortions. With appropriate $\gamma$, a small $\alpha$ leads to texture distortion. A large $\alpha$ leads to smoothing and hazy appearance. By carefully tuning $\alpha$, the details can be adjusted for balance. Then, the decomposition results with varying $\beta$ are shown in Fig. 6b. When $\beta$ is small, the constraint on $\mathbf{S}$ is weak and many contents are mistakenly decomposed into $\mathbf{S}$. When $\beta$ is large enough, $\mathbf{S}$ contains little information, resulting in residual noise in $\mathbf{L}$.

**Ablation Study.** We validate some architecture designs and loss functions. For architectures, we consider: i) intra-modal recovery in I2Former. We remove it and modify I2Former to a Restormer-like structure with similar parameters. ii) direct fusion. Rather than learning a residual image to noisy inputs, I2Former directly generates fused image. It learns a residual image $\mathbf{F}_r$ to a pre-fused image $\mathbf{F}_p$ as fused image $\mathbf{F} = \mathbf{F}_r + \mathbf{F}_p$, where $\mathbf{F}_p = c(\max(\mathbf{V}^y, \mathbf{R}), \mathbf{V}^{Cb}, \mathbf{V}^{Cr})$. For loss functions, we remove the gradient loss $\mathcal{L}_g$ and regularization term $\mathcal{L}_{rank}$. As in Fig. 6c, $\mathbf{F}_p$ contains noise. Without intra-modal recovery, the result shows diagonal effects. If learning a residual image to $\mathbf{F}_p$, the unstable training process results in slight intensity and color casts. The lack of $\mathcal{L}_g$ results in blur contents. Without $\mathcal{L}_{rank}$, the result cannot infer more contents according to convolutional low-rank property. Deno-IF generates clear results with more contents. The results in Tab. 2a are also consistent with qualitative results. Without $\mathcal{L}_g$, it shows the worst performance. By introducing intra-modal recovery, direct fusion, and $\mathcal{L}_{rank}$, our method achieves the optimal performance among all the settings.

**Performances on Different-Level Noise**. Experiments on different-level noise Gaussian noise (as a representative) are reported in Tab. 2b. As $\sigma$ increases, the performances of the competitors significantly decrease on most metrics. Although it is an increasingly challenging trend for recovering under more severe noise, our method shows smaller fluctuations, showing its robustness.

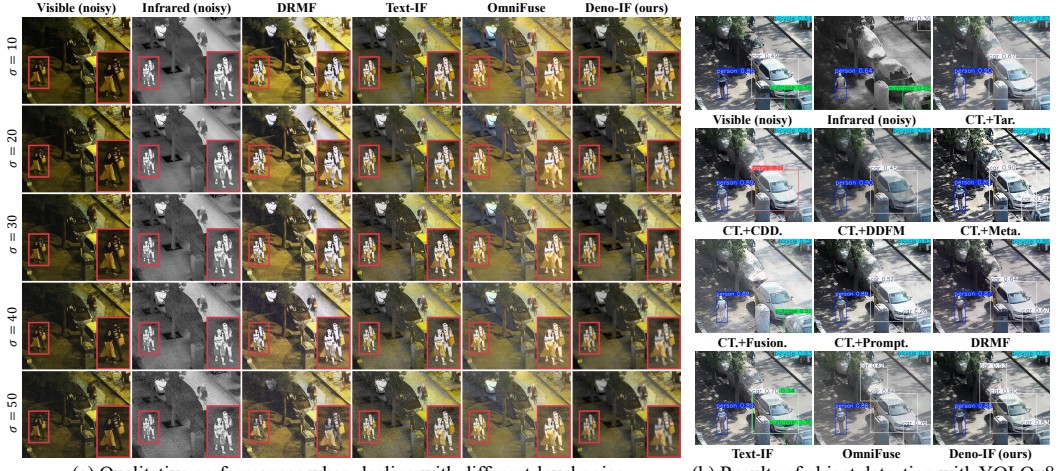

(a) Qualitative performances when dealing with different-level noise.  (b) Results of object detection with YOLOv8.

Figure 7: Results of different-level noise and external verification on high-level task.

Table 3: Computational efficiency comparison of different methods.

| Methods | Tar. | CDD. | DDFM | Meta. | Fusion. | Prompt. | CT.+Tar. | CT.+CDD. | CT.+DDFM | CT.+Meta. | CT.+Fusion. | CT.+Prompt. | DRMF | Text-IF | Omn. | Deno-IF |
|---|---|---|---|---|---|---|---|---|---|---|---|---|---|---|---|---|
| **Para (M)** | 0.30 | 1.19 | 552.66 | 0.81 | 0.56 | 7.44 | 51.71 | 52.60 | 604.07 | 52.22 | 51.97 | 58.91 | 170.98 | 215.12 | 98.15 | **1.433** |
| **FLOPs (G)** | 91 | 548 | 5221 | 159 | 241 | 36877 | 41695 | 42152 | 46824 | 41763 | 41845 | 78481 | 4576 | 1519 | 5471 | **115** |
| **Runtime (s)** | 0.03 | 0.23 | 68.79 | 0.04 | 2.49 | 1.08 | 13.50 | 13.70 | 82.26 | 13.51 | 15.96 | 14.55 | 4.47 | 0.31 | 4.53 | **0.06** |

**External Verification on High-Level Task.** The denoising and fusion performances are externally validated by a subsequent high-level task–object detection. In Fig. 7b, it is challenging to accurately detect some targets in a single noisy visible/infrared image. By denoising and fusing noisy source images, some competitors can boost detection accuracy while some methods exhibit notable misclassification. Our result eliminates noise interference and preserve essential information, improving both detection accuracy and precision. Quantitative results are reported in *Supplementary Material*.

**Efficiency Comparison**. The efficiency is evaluated by parameter numbers, FLOPs, and runtime. Due to the large parameters, high computational complexity, and runtime of denoising method CTNet, we also report the efficiency of pure fusion methods. As reported in Tab. 3, our method shows higher efficiency than the competitors. It also achieves simultaneous denoising and fusion with an efficiency that is comparable or even superior to those of pure fusion networks.

## 5 Conclusion

This paper proposes an unsupervised noisy visible and infrared image fusion method, addressing the limitations of existing methods that cannot handle noise interference or rely on paired data restricted by noise characteristics. The proposed network, featuring a convolutional low-rank optimization module and a joint denoising and fusion module, eliminates data dependency and improves generalization while reducing complexity. Experimental results demonstrate its effectiveness and robustness.

## Acknowledgments and Disclosure of Funding

This work was supported by the National Natural Science Foundation of China (NSFC) under Grant U21B2027 and 62502085, the Natural Science Foundation of Jiangsu Province of China under Grant BK20241280, and the Start-up Research Fund of Southeast University under Grant RF1028623006 and RF1028624061.

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

# Technical Appendices and Supplementary Material

## 1. Details of Transformer Block and Variation of Losses under Different Network Settings

The network architecture of the Transformer Block in Fig. 2 is shown in Fig. 8a, where the left part is multi-dconv head transposed attention and the right part is the gated dconv feed-forward network.

In the ablation study introduced in Sec. 4, we perform the experiments by removing the intra-modal recovery and learning a residual image to a pre-fused image (w/o direct fusion). The variation of losses under these settings during the training phase is shown in Fig. 8b. By comparison, the I2Former in Deno-IF achieves both faster convergence speed and higher convergence accuracy.

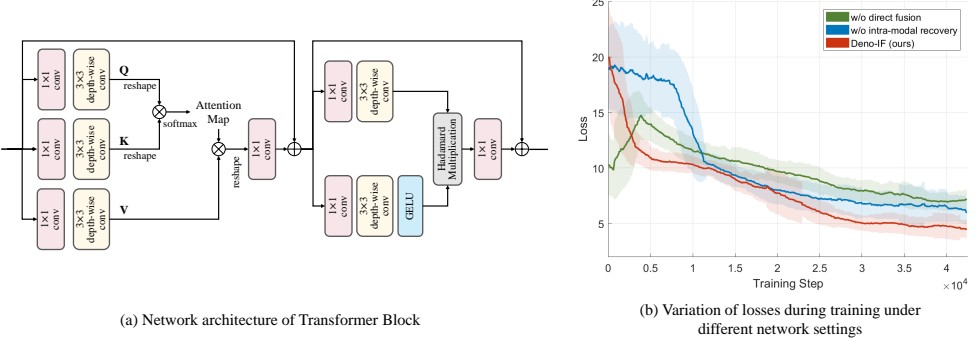

(a) Network architecture of Transformer Block      (b) Variation of losses during training under different network settings

Figure 8: Details of Transformer block and variation of losses under different network settings.

## 2. Quantitative Results of Generalization in More Situations of Noise Presence

The quantitative results on five metrics across two datasets when fusing i) noisy visible and clean infrared images; ii) clean visible and noisy infrared images are reported in Tabs. 4 and 5, respectively. In these situations, our method can still achieve optimal or comparable performances, indicating its generalization for various situation of noise presence.

Table 4: Quantitative results when fusing noisy visible and clean infrared images (**bold**: optimal, underline: suboptimal).

| Datasets | LLVIP | | | | | M3FD | | | | |
|---|---|---|---|---|---|---|---|---|---|---|
| Metrics | SSIM↑ | PSNR↑ | FSIM↑ | CC↑ | BRISQUE↓ | SSIM↑ | PSNR↑ | FSIM↑ | CC↑ | BRISQUE↓ |
| CT.+Tar. | 0.370±0.133 | 15.174±1.444 | 0.804±0.028 | 0.691±0.073 | 25.623±8.442 | 0.683±0.111 | 15.423±2.617 | 0.809±0.034 | 0.532±0.199 | 21.485±5.250 |
| CT.+CDD. | 0.459±0.150 | 15.665±1.510 | 0.814±0.026 | 0.712±0.068 | 37.863±19.525 | 0.674±0.125 | 15.609±2.704 | 0.812±0.043 | 0.520±0.224 | 27.945±6.977 |
| CT.+DDFM | 0.493±0.112 | 16.915±1.353 | 0.825±0.020 | **0.753±0.068** | 32.374±15.354 | **0.699±0.111** | 15.929±2.368 | 0.814±0.056 | **0.612±0.189** | 29.635±6.150 |
| CT.+Meta. | 0.494±0.072 | 15.348±1.285 | 0.687±0.033 | 0.687±0.078 | 30.694±7.736 | 0.642±0.114 | 14.926±2.402 | 0.729±0.052 | 0.569±0.185 | **16.723±9.889** |
| CT.+Fusion. | 0.309±0.100 | 11.756±0.663 | 0.816±0.020 | 0.720±0.075 | 36.910±13.231 | 0.636±0.075 | 10.951±0.983 | **0.824±0.038** | 0.605±0.151 | 26.701±7.944 |
| CT.+Prompt. | 0.449±0.145 | 15.743±1.410 | 0.814±0.025 | 0.713±0.069 | 37.690±17.569 | 0.659±0.128 | 15.561±2.593 | 0.806±0.045 | 0.474±0.243 | 26.715±7.489 |
| DRMF | 0.116±0.045 | 11.124±0.688 | 0.673±0.019 | 0.489±0.150 | 75.159±4.613 | 0.346±0.144 | 13.752±2.144 | 0.739±0.043 | 0.351±0.232 | 42.136±15.254 |
| Text-IF | 0.365±0.112 | 14.571±1.222 | 0.776±0.022 | 0.696±0.071 | **15.487±8.137** | 0.660±0.137 | 15.400±3.025 | 0.814±0.044 | 0.465±0.225 | 38.897±8.459 |
| OmniFuse | 0.456±0.061 | 12.252±0.852 | 0.748±0.014 | 0.654±0.093 | 40.314±2.856 | 0.663±0.101 | 15.255±2.486 | 0.801±0.033 | 0.439±0.237 | 53.264±5.029 |
| Deno-IF | **0.685±0.040** | **17.104±1.691** | **0.827±0.022** | 0.749±0.076 | 23.779±4.270 | 0.691±0.082 | **15.973±2.905** | 0.822±0.032 | 0.592±0.174 | 27.164±5.110 |

Table 5: Quantitative results when fusing clean visible and noisy infrared images.

| Datasets | LLVIP | | | | | M3FD | | | | |
|---|---|---|---|---|---|---|---|---|---|---|
| Metrics | SSIM↑ | PSNR↑ | FSIM↑ | CC↑ | BRISQUE↓ | SSIM↑ | PSNR↑ | FSIM↑ | CC↑ | BRISQUE↓ |
| CT.+Tar. | 0.341±0.138 | 14.610±1.300 | 0.772±0.021 | 0.687±0.077 | 20.811±8.946 | 0.511±0.177 | 15.467±2.243 | 0.778±0.036 | 0.609±0.161 | 19.693±9.830 |
| CT.+CDD. | 0.346±0.154 | 14.992±1.316 | 0.785±0.021 | 0.709±0.073 | 44.857±11.512 | 0.541±0.165 | 16.048±2.466 | 0.798±0.026 | 0.605±0.188 | 35.030±17.913 |
| CT.+DDFM | 0.419±0.137 | 16.064±1.220 | **0.803±0.017** | **0.752±0.073** | 36.578±11.318 | 0.555±0.177 | 16.211±2.278 | 0.803±0.042 | **0.675±0.148** | 30.017±15.380 |
| CT.+Meta. | 0.543±0.060 | 14.701±1.189 | 0.685±0.034 | 0.676±0.084 | **18.320±10.509** | 0.608±0.100 | 15.512±2.471 | 0.696±0.043 | 0.642±0.150 | **10.293±7.004** |
| CT.+Fusion. | 0.442±0.062 | 11.516±0.652 | 0.757±0.029 | 0.742±0.072 | 31.142±5.761 | 0.573±0.083 | 11.081±0.835 | 0.784±0.037 | 0.663±0.130 | 18.544±5.514 |
| CT.+Prompt. | 0.332±0.146 | 14.845±1.311 | 0.782±0.022 | 0.707±0.073 | 47.847±12.074 | 0.550±0.143 | 16.326±2.311 | 0.795±0.026 | 0.576±0.208 | 25.187±14.365 |
| DRMF | 0.560±0.055 | 13.508±1.760 | 0.739±0.027 | 0.601±0.133 | 25.842±7.661 | **0.693±0.078** | 15.675±2.150 | 0.808±0.026 | 0.452±0.221 | 46.885±10.449 |
| Text-IF | 0.546±0.089 | 15.097±1.355 | 0.770±0.026 | 0.697±0.066 | 22.293±7.881 | 0.660±0.113 | 16.505±2.464 | 0.795±0.032 | 0.561±0.211 | 25.388±6.248 |
| OmniFuse | 0.548±0.050 | 12.721±0.912 | 0.754±0.021 | 0.688±0.076 | 36.807±4.436 | 0.688±0.091 | 15.790±2.374 | 0.806±0.023 | 0.541±0.215 | 48.140±4.642 |
| Deno-IF | **0.660±0.052** | **16.072±1.183** | 0.797±0.017 | 0.732±0.064 | 20.248±7.930 | 0.684±0.072 | **16.530±2.496** | **0.812±0.023** | 0.650±0.172 | 19.422±9.141 |

## 3. Results on the Original Clean Visible and Infrared Image Fusion Task

We conduct the experiment to fuse the original clean visible and infrared images. The experiment merely compares the fusion performances of Deno-IF and SOTA image fusion methods and verifies whether the proposed method will blur clean data. As shown in Figs. 9-10, our method can preserve the content of two source images in a more balanced manner, thus reflecting more scene contents in the fused image and assisting in human visual perception with comparable contrast.

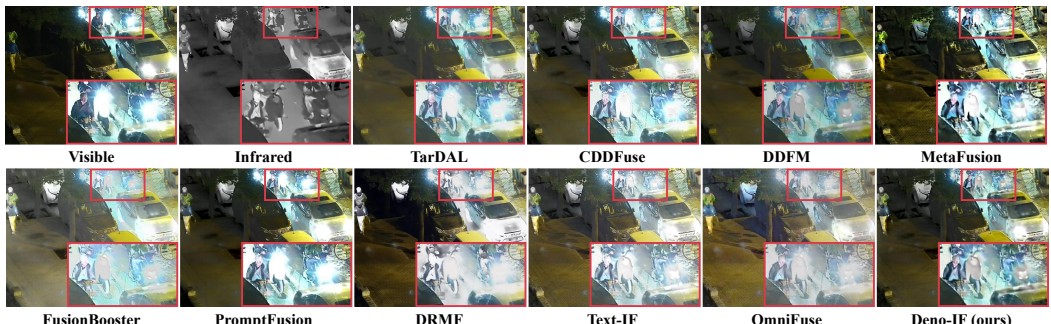

Figure 9: Qualitative results on the LLVIP dataset when fusing clean visible and infrared images.

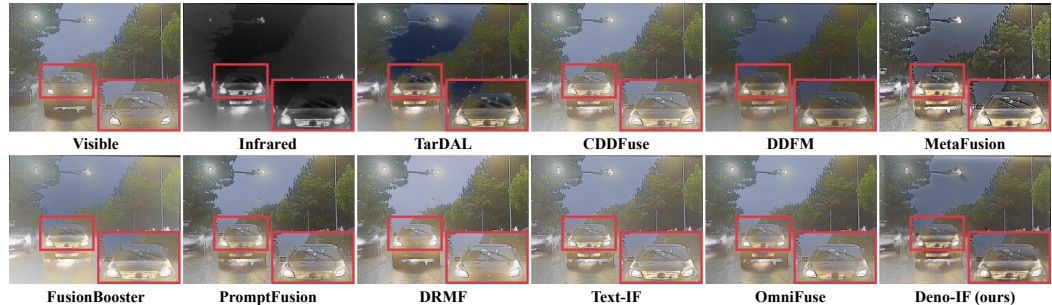

Figure 10: Qualitative results on the M3FD dataset when fusing clean visible and infrared images.

## 4. Quantitative Results of Detection under Noise Interference

Tab. 6 reports the quantitative results of detection on the fusion results of noisy source images. The metrics include precision, recall, average precision (AP), and mean AP (mAP). mAP is the average of all APs at IoU thresholds from 0.5 to 0.95 in steps of 0.05.

Table 6: Object detection results when source images suffer from noise.

|  | CT.+Tar. | CT.+CDD. | CT.+DDFM | CT.+Meta. | CT.+Fusion. | CT.+Prompt. | DRMF | Text-IF | OmniFuse | Deno-IF |
|---|---|---|---|---|---|---|---|---|---|---|
| Precision | 0.888 | 0.917 | 0.877 | **0.917** | 0.857 | 0.883 | 0.877 | 0.902 | 0.897 | 0.878 |
| Recall | 0.619 | 0.670 | 0.701 | 0.653 | 0.657 | 0.645 | 0.601 | 0.688 | 0.697 | **0.724** |
| AP@0.5 | 0.763 | 0.787 | **0.811** | 0.772 | 0.758 | 0.769 | 0.722 | 0.786 | 0.802 | 0.807 |
| mAP | 0.488 | 0.516 | 0.535 | 0.496 | 0.489 | 0.496 | 0.437 | 0.507 | 0.525 | **0.536** |

## 5. Validation of Convergence during Iterative Process

The first module is the convolutional low-rank module, which aims to decompose the noisy input into clean and noisy components. The decomposition problem is rewritten as the energy function in Eq. (5). To validate the convergence during the iterative process, we document several terms of this energy function during iteration.

As summarized in Tab. 7, the iterative process of low-rank optimization module demonstrates convergence. For **M** and **N**, they are initialized as rectangular identity-like matrices (all elements

on the main diagonal are 1 and other elements are 0). As the initialized matrices do not contain much information, $\|\mathbf{M}\|_F$ and $\|\mathbf{N}\|_F$ initially increase to incorporate more information, followed by a gradual stabilization as iteration progresses.

Table 7: Changes of various terms in the energy function during the iteration process.

| Iterations | 2 | 4 | 6 | 8 | 10 | 12 | 14 | 16 |
|---|---|---|---|---|---|---|---|---|
| $\|\mathbf{R} - \mathbf{L} - \mathbf{S}\|_F$ | 227.970 | 87.555 | 44.188 | 33.726 | 31.791 | 31.446 | 31.390 | 31.390 |
| $\|\mathcal{A}_k(\mathbf{L}) - \mathbf{MN}\|_F$ | 3821.812 | 2395.634 | 1220.628 | 940.686 | 889.409 | 880.317 | 878.878 | 878.862 |
| $\|\mathbf{S}\|_F$ | 0.456 | 0.175 | 0.088 | 0.067 | 0.064 | 0.063 | 0.063 | 0.063 |

From the aspect of network optimization, clean components derived with convolutional low-rank priors serve as i) physics-driven prior injector to regularize fused images, and ii) implicit teachers that enable knowledge distillation of denoising priors into the fusion network. It enables simultaneous noise suppression and information fusion

## 6. Analysis of Computational Training Cost

As the training latency is dominated by the first stage (convolutional low-rank optimization module), we quantify the computational bottleneck within this module. In this module, computational complexity scales primarily with i) convolution kernel size ($k_1, k_2$) and ii) number of iterations ($T$) for solving the low-rank subproblem. The training costs of different settings are reported in Tab. 8. It is observed that increasing $k_1, k_2$ incurs significant increase in training time while the number of iteration scales linearly with computational cost. Nevertheless, the entire training process remains practical, completing within several days.

Table 8: Changes of various terms in the energy function during the iteration process.

| $k_1, k_2$ | 12 | 18 | 24 | 30 | 36 | 42 |
|---|---|---|---|---|---|---|
| Training Time (m) / Epoch | 41.31 | 54.35 | 71.51 | 93.28 | 121.22 | 151.68 |
| $T$ | 10 | 20 | 30 | 40 | 50 | 60 |
| Training Time (m) / Epoch | 20.58 | 30.85 | 41.31 | 51.61 | 61.62 | 71.62 |

## 7. Differences with Prior Low-Rank Methods in Other Domains

While low-rank methods exist in some image processing (e.g., image smoothing, restoration, denoising), the proposed method differs from existing works in task formulation and low-rank theory.

For task formulation, the proposed method differs with prior low-rank methods in input complexity and objective shift. For input complexity, we handle multi-modal data rather than single-modal inputs. For objective, we focus on joint denoising and fusion rather than a single image processing task.

For low-rank theory, the proposed theory differs in energy function formulation and optimization strategy. For energy function, the regularization term is based on convolutional low-rank prior by the convolution nuclear norm. Compared with standard low-rank theory, it can avoid excessive smoothing and structural damage for better local structure preservation, noise robustness, and physical interpretability. For optimization strategy, previous researches directly implement the objective function as the loss of network. Differently, we design a two-phase optimization strategy. In the first phase, we rewrite the energy function and iteratively address the subproblems with analytical solutions. In the second phase, we design a deep network to distill the physics-driven priors and realize denoising and fusion jointly.

## 8. Limitations

The convolutional low-rank optimization module serves as the theoretical cornerstone of the joint denoising and fusion network. It ensures physically-consistent guidance through iterative refinement during training but requires iterative optimization. Although it is not applied in the testing phase, it lengthens the overall training time. In future work, we plan to investigate more efficient optimization schemes to preserve performance while reducing training costs.

