# OpenReview forum: "Deno-IF: Unsupervised Noisy Visible and Infrared Image Fusion Method"
_NeurIPS.cc/2025/Conference — NeurIPS 2025 spotlight_

### Official Review · Reviewer_XjA5 · 2025-06-05

**Clarity:** 4
**Significance:** 3
**Originality:** 3
**Rating:** 5
**Confidence:** 4

**Summary:**

This paper explores how noise degradation affects image fusion in real-world imaging scenarios and introduces Deno-IF, an unsupervised method for fusing noisy visible and infrared images. Leveraging the convolutional low-rank priors of clean images, it first develops a convolutional low-rank optimization module to extract clean image components from noisy inputs. Following this, a specially designed network, combining inter-modal recovery and intra-modal processing, performs denoising and fusion simultaneously. As a result, the proposed method achieves effective noise-suppressed image fusion with fewer network parameters. Extensive experiments are conducted under various noise types, intensities, and presence conditions to assess the method’s effectiveness and generalization capability.

**Questions:**

i) To assess robustness, the experiments involve various levels of Gaussian noise. However, it is unclear whether this noise is added to the visible image, the infrared image, or both.

ii) The parameters in the convolutional low-rank optimization module are determined using manually designed operations aimed at estimating the noise level. What is the rationale behind this choice?

iii) The linear mapping $A_k(\cdot)$ transforms an image of size $h \times w$ into a convolutional matrix of size $hw \times k_1k_2$. Has there been any investigation into whether increasing $k_1$ and $k_2$ yields improvements in performance?

iv) The final loss function includes several hyperparameters. While most are kept constant during training, $\kappa$ is gradually increased. Why?

**Ethical Concerns:**

["NO or VERY MINOR ethics concerns only"]

**Final Justification:**

Thank the authors for their efforts in clarifying these points. As a result, the issues have been addressed, and I will maintain my rating.

**Limitations:**

Yes.

**Paper Formatting Concerns:**

None.

**Quality:**

4

**Strengths And Weaknesses:**

Strengths

i) The work addresses the challenge of noisy data in real-world conditions by proposing a unified framework that integrates denoising and fusion, making it suitable for a broader range of practical applications.

ii) The proposed approach is fully unsupervised—it eliminates the need for paired noisy and clean image data by automatically estimating clean components using convolutional low-rank decomposition.

iii) The method not only applies the convolutional low-rank prior in the decomposition stage but also introduces it as part of the loss function to enhance the quality of the fused image. This represents a novel loss formulation in the context of image fusion.

iv)The paper presents its methodology with clear structure, and the experimental evaluations are thorough—covering diverse noise types and levels—which demonstrates both the effectiveness and strong generalization ability of the proposed method.

Weaknesses

i) The training process of the proposed method is both complex and computationally intensive. This is largely due to the iterative nature of the convolutional low-rank optimization, which is further burdened by the need to train the denoising and fusion components of the network jointly.

ii) The paper does not sufficiently explain how the no-reference metric BRISQUE is used to evaluate the quality of the fused images. Additionally, the results of BRISQUE assessments under various noise levels should be included to enhance the evaluation.

iii) There are several typographical issues that need correction: (a) In the abstract, the phrase “intra-modal recovery and intra-modal recovery and fusion” should be revised to “intra-modal recovery and inter-modal recovery and fusion.” (b) The expression “Experiments on different-level noise Gaussian noise” is redundant and unclear, and should be corrected for clarity.

---

> ### Author Rebuttal · Authors · 2025-07-31
>
> >**Q1: Training complicacy and computation complexity.**
>
> **A1**: We appreciate this insightful observation. To enhance methodological clarity, the final version will include a detailed algorithmic presentation in pseudocode format.
>
> Besides, we want to clarify the computational efficiency in the testing phase. While the iterative optimization in the convolutional low-rank module increases computational overhead, the complexity is confined to the training phase. This module serves solely as an implicit knowledge distillation mechanism to guide the joint denoising and fusion network. During inference, it is completely bypassed, not incurring additional computation. The computational efficiency comparison results in Tab. 3 in the submitted manuscript also indicate that our method achieves optimal efficiency (including parameter numbers, FLOPs, and inference time) during the testing phase.
>
> >**Q2: No-reference metric BRISQUE and no-reference results under different-level noise.**
>
> **A2**: BRISQUE measures image quality by quantifying deviations from natural scene statistics in spatial domain. It extracts spatial features from the empirical distribution of locally normalized luminance and products of locally normalized luminance under a spatial natural scene statistic model. The features capture losses in naturalness caused by distortions, generating a holistic measure of quality. The no-reference comparison results under multi-level noise are reported in the following table. Key observations are analyzed from two perspectives:
>
> |$\sigma$|10|20|30|40|50|
> |--|--|--|--|--|--|
> |DRMF|26.855|58.717|66.016|74.548|75.814|
> |Text-IF|33.060|45.843|58.087|61.441|66.258|
> |Deno-IF|15.364|16.493|17.989|20.514|24.256|
>
> 1. Robustness to increasing noise severity
>
> As noise level escalates, comparison degradation-aware image fusion methods exhibit significant performance degradation, while our method maintains stable quality preservation. It demonstrates our method’s inherent robustness to multi-level noise interference.
>
> 2. Effectiveness per noise level
>
> At each noise level, our method achieves best-in-class BRISQUE scores, outperforming comparison degradation-aware image fusion methods. The consistently low BRISQUE scores confirm our method's robustness in joint denoising-fusion, effectively maintaining naturalness under noise interference.
>
> We will add the introduction and selection reasons of BRISQUE and other metrics and provide the above results in the final version.
>
> >**Q3: Clarification on multi-level Gaussian noise.**
>
> **A3**: Thanks for the query about multi-level Gaussian noise targets. To validate the robustness, we perform the experiment where both the visible and infrared images suffer from gaussian of different-level Gaussian noise. We will provide qualitative results of multi-level noise in the final version, which demonstrates our method’s robustness to multi-level noise interference.
>
> >**Q4: Parameter settings in the convolutional low-rank optimization module.**
>
> **A4**: These settings stem from principled trade-offs and computational efficiency:
>
> 1. Principled trade-off
>
> These parameters critically balance three competing objectives, including data fidelity, regularization, and releasing terms. The parameter settings directly correlate with the characteristics of source images, particularly noise levels. Higher noise level demands stronger regularization to suppress noise while lower noise level requires weaker regularization to preserve details.
>
> 2. Computational efficiency imperative
>
> While deep learning-based noise estimation networks exist, such approaches incur prohibitive computational overhead, especially when the training process is already time-consuming. The manually designed estimator can achieve optimal efficiency-accuracy trade-off.
>
> Considering these two aspects, a compromise is made by using manual operations. We will add these reasons of manually designed operations for parameter settings in the final version.
>
> >**Q5: Does increasing $k_1,k_2$ lead to performance improvements?**
>
> **A5**: Theoretically, increasing the size of convolutional kernel can contain more structural information for inference. However, excessively large convolutional kernels constrain the convolutional low-rank nature of more complex scenes, which may potentially comprise the preservation of complex information.
>
> In terms of experimental verification, we increase $k_1,k_2$ and retrain the network. As empirically validated in the table below, appropriately increasing $k_1,k_2$ improves the performance. When $k_1,k_2$ may be excessively (due to time and equipment limitations, we increase $k_1,k_2$ to 36 for feasible experiments to observe trends), enforcing convolutional low-rank constraints over largely expansive regions risks damaging local details. It is reflected in a slightly decreasing performance trend.
>
> |$k_1,k_2$|12|18|30|36|
> |--|--|--|--|--|
> |Training time/epoch (m)|**41.31**|$\underline{54.35}$|93.28|121.22|
> |SSIM$\uparrow$|0.616|0.617|$\underline{0.640}$|**0.643**|
> |PSNR$\uparrow$|17.070|17.122|**17.286**|$\underline{17.239}$|
> |FSIM$\uparrow$|0.811|0.814|$\underline{0.818}$|**0.819**|
> |CC$\uparrow$|0.738|0.733|**0.737**|$\underline{0.736}$|
> |BRISQUE$\downarrow$|16.725|$\underline{16.633}$|**16.155**|18.713|
>
> Besides, computationally, increasing kernel sizes $\{k_1,k_2\}$ generates a convolutional matrix of dimensions $ hw\times{k_1k_2}$, leading to prohibitive computational costs.
>
> Thanks again for this question which provides more validation and optimized settings of $k_1,k_2$. We will add the experiments about different settings of $k_1,k_2$  in the final version.
>
>
>
> >**Q6: Progressively increasing setting of $\kappa$.**
>
> **A6**: The dynamically increasing setting of $\kappa$ is motivated by the need to balance competing objectives: data fidelity (similarity losses between source and fused data) and regularization (convolutional low rankness of fused images). Through extensive experiment, we observe that: i) Large initial $\kappa$ forces the network to over-prioritize the low-rank constraint, causing fused image to be excessively simplified (overly low-rank). It suppresses critical structural details in source images, leading to serious information distortion. ii) Small initial $\kappa$ allows data fidelity to dominate early training, progressively incorporating richer information into fused image. However, a fixed small $\kappa$ later renders the regularization ineffective, failing to refine high-frequency details. The scheduled increasing strategy of $\kappa$ prioritizes data fidelity to preserve scene content in the early stage and gradually strengthen the low-rank prior to refine textures in the later stage.
>
> >**Q7: Typographical issues to be corrected.**
>
> **A7**: Thanks for point out the typos. We will carefully correct them as “intra-modal recovery and inter-modal recovery and fusion” and “different-level Gaussian noise” in the final version and further improve the description.

---

### Official Review · Reviewer_oJaz · 2025-06-19

**Clarity:** 3
**Significance:** 4
**Originality:** 3
**Rating:** 5
**Confidence:** 5

**Summary:**

This paper proposes Deno-IF, an unsupervised method for fusing noisy visible-infrared images that simultaneously addresses denoising and fusion without requiring clean training data. The key innovation lies in two synergistic modules: (1) A convolutional low-rank optimization module that decomposes noisy inputs into clean components through nuclear norm minimization, providing physics-guided training signals; and (2) I2Former, a transformer network combining intra-modal recovery (self-attention denoising) and inter-modal fusion (cross-attention integration). Extensive experiments on M3FD and LLVIP demonstrate that the proposed method exhibits advantageous capabilities in denoising and aggregating complementary information, along with strong generalization.

**Questions:**

1. According to the parameters of most comparison methods, the parameter of CTNet is about 50M, while the parameter of SwinFusion+CTNet is 5.38M.

2. In the joint denoising and fusion module, there are many transformer blocks either for inter-modal recovery or intra-modal recovery and fusion. The features of these blocks are different. Can Eqs. (10)-(11) be distinguished based on different blocks?

3. Why is CTNet chosen as the comparison method for denoising source images instead of other methods?

**Ethical Concerns:**

["NO or VERY MINOR ethics concerns only"]

**Final Justification:**

The response effectively addressed my concerns, and I have decided to maintain my accept score.

**Limitations:**

Yes

**Paper Formatting Concerns:**

N / A

**Quality:**

4

**Strengths And Weaknesses:**

## Strengths

1. This manuscript proposes an unsupervised image fusion network for noisy infrared and visible images, which fully exploits the low-rank characteristics of clean data.

2. By integrating the proposed convolutional low-rank optimization module, the authors design a lightweight joint denoising and fusion module that maintains strong denoising and information integration capabilities while ensuring high computational efficiency.

3. Extensive comparative experiments, ablation studies, downstream task evaluations, and efficiency analyses validate the effectiveness of the proposed approach.

4. Targeting the noise issues commonly encountered in real-world infrared and visible image fusion systems, the proposed method contributes to broadening the application scope of such fusion tasks. Moreover, the paper declares clear motivation and well-organized writing, which is easy to follow and understand.

## Weaknesses

1. The Convolutional Low-Rank Optimization Module and the Joint Denoising and Fusion Module appear somewhat decoupled. Emphasizing their connection would help readers better understand the proposed framework.

2. The related work discusses existing image fusion methods. Highlighting how this method differs from prior low-rank methods in other domains will further improve the contribution of this work.

3. In Figure 1, the noise in the visible image is not visually obvious. Replacing it with a scene where the noise is more clearly noticeable could better highlight the value of the proposed method. Besides, some details need to be corrected. For example, an extra “.” appears on line 60, and the acronym MAP on line 128 should be explicitly explained.

---

> ### Author Rebuttal · Authors · 2025-07-30
>
> >**Q1: Connection between two modules.**
>
> **A1**: We sincerely appreciate this for this constructive suggestion. We want to emphasize that the connection between these modules operates not through superficial explicit-level structural transfer of data or feature, but fundamentally via deep implicit physics-driven constraint propagation. The ultimate goal is to obtain a joint denoising and fusion module via an unsupervised approach, yet struggles to infer high-quality structural cues to guide network optimization. To address it, the convolutional low-rank optimization module leverages convolutional low-rank priors to derive clean components. These components serve as i) physics-driven prior injector to regularize fused images, and ii) implicit teachers that enable knowledge distillation of denoising priors directly into the final joint denoising and fusion network. The design enables simultaneous noise suppression and information fusion while eliminating the need for low-rank module reuse during testing, thereby enhancing real-time inference efficiency. We will emphasize the deep implicit connection in the final version.
>
> >**Q2: Differences with prior low-rank methods in other domains.**
>
> **A2**: While low-rank methods exist in image processing (e.g., image smoothing, image restoration, image denoising), the proposed method differs from existing works in both task formulation and low-rank theory:
>
> 1. Task formulation
>
> The proposed method differs in input complexity and objective shift. For input complexity, we handle multi-modal data rather than single-modal inputs. For objective, we focus on joint denoising and fusion rather than a single image processing task.
>
> 2. Low-rank theory
>
> The proposed theory differs in energy function formulation and optimization strategy. For energy function, the regularization term is based on convolutional low-rank prior by the convolution nuclear norm. Compared with standard low-rank theory, it can avoid excessive smoothing and structural damage for better local structure preservation, noise robustness, and physical interpretability. For optimization strategy, previous researches directly implement the objective function as the loss of network. Differently, we design a two-phase optimization strategy. In the first phase, we rewrite the energy function and iteratively address the subproblems with analytical solutions. In the second phase, we design a deep network to distill the physics-driven priors and realize denoising and fusion jointly.
>
> We will highlight the contributions of the proposed method over existing low-rank methods in other domains in the final version.
>
> >**Q3: Replace Fig. 1 with more noticeable noise.**
>
> **A3**: We appreciate this insightful recommendation. We will replace the qualitative examples in Fig.1 with the comparison examples where both the visible and infrared images suffer from more clearly noticeable noise in the final version. Compared with the combination of SOTA image denoising and fusion methods, the proposed method can restore and represent essential high-quality structural information from source images with greater clarity in the fusion result.
>
> >**Q4: Annotate the block number in equations.**
>
> **A4**: Thanks for this suggestion. In both intra-modal recovery and inter-modal recovery and fusion parts, there are many blocks. To avoid the confusion between the block numbers of these two parts, we discuss them separately. In Eq. (10) for intra-modal recovery, the output of the current block (e.g., $\hat{\textbf{f}}_v$) is only related to the input of the current block (e.g., $\textbf{f}_v$). It does not involve complex update process as Eqs. (6a)-(6d). For the sake of formula simplicity, the block numbers will not be given. In Eq. (11) for inter-modal recovery and fusion, the input of each block includes visible, infrared and even earlier fused features. However, the directions of intra-modal recovery and inter-modal recovery and fusion are different. The earlier blocks of inter-modal recovery and fusion need to fuse the features of the later blocks of intra-modal recovery. This is clearly represented in Fig. 2, so the block number will not be labeled to avoid confusion.
>
>
> >**Q5: Reason of selecting CTNet for comparison.**
>
> **A5**: The rationale of selecting CTNet for comparison is two-fold: generalization validation between unsupervised and supervised approaches, and architecturally relevant comparison.
>
> 1. Validating unsupervised generalization against supervised performance
>
> CTNet represents a SOTA supervised denoising approach, requiring paired clean and noisy images for training. Our method is an unsupervised denoising and fusion framework, leveraging the convolutional low-rank prior without needing clean data. Including CTNet for comparison serves a critical purpose: testing the generalization and denoising effectiveness of our unsupervised approach. Comparing against a top-performing supervised denoising method, our unsupervised method achieves competitive or even superior denoising performance. It validates the robustness and applicability of our approach, especially when paired training data is unavailable or difficult to obtain.
>
> 2. Architecturally relevant comparison
>
> CTNet and our method are fundamentally based on Transformer. By comparing with CTNet, it provides a controlled comparison point, specifically highlighting improvements of divergent frameworks and objectives. CTNet utilizes Transformer primarily to mine structural information and explore multi-level interactions for extra information, all within a single modality for denoising. In contrast, our method uses Transformer for both multi-modal image denoising and fusion. It simultaneously addresses intra-modal recovery and inter-modal recovery and fusion within a unified framework. Comparing against CTNet, a typical representative of Transformer-based denoisers, allows to directly highlight the effectiveness and novelty of our multi-modal denoising- and fusion-oriented Transformer design.
>
> We will provide a brief introduction to CTNet and the reason for leveraging it for comparison in the final version.
>
> >**Q6: Typo and description.**
>
> **A6**: Thanks for pointing out the issues related to typo and detailed description. The parameter of SwinFusion+CTNet is 52.38M. The circle size in Fig.1 represents the correct parameter number. MAP represents classic Maximum A Posteriori inference to integrate physics-driven priors with data-driven fitting for robust decomposition. We will carefully correct the typos (extra “.” and parameter) and description in the final version and further correct the presentation of figures and tables.

---

> > ### Comment · Reviewer_oJaz · 2025-08-04
> >
> > Thank you for the rebuttal. It effectively addressed my concerns, and I have decided to maintain my accept score.

---

### Official Review · Reviewer_GvvZ · 2025-07-01

**Clarity:** 2
**Significance:** 2
**Originality:** 3
**Rating:** 4
**Confidence:** 5

**Summary:**

This paper proposed an unsupervised method to fuse noisy infrared and visible images. The proposed method consists of two parts. The first part is a convolutional low-rank optimization module, which is used to generat clean images. The second part is a joint denoising and fusion module. The main advantage of the proposed method is that it is unsupervised. Additionally, a convolution matrix-based regularization loss is designed. A series of experiments on simulated noisy dataset are conducted to demonstrate the effectiveness of the proposed method.

**Questions:**

1. What is the motivation of this work? Based on the reviewer's experience, images collected using infrared an visible cameras are usually not noisy. Especially, the infrared images captured using a thermal infrared camera are clean.  The reviewer does not see the need of fusing noisy images, especially the manually-created noisy images. More convincing details should be provided to explain the motivcation. Otherwise, it seems that the authors created an issue (that may not exist in practise) and then proposed a method to solve this issue.

2. Does the convolutional low-rank optimization module like a pre-processing module? The clean data generated by this module is then used as label for the supervision of the joint denosing and fusion module. Is it possible to replace this convolutional low-rank optimization module using an unsupervised denoising model, and then used the denoised data as supervision for the joint denosing and fusion module? What is the disadvantage of using an unsupervised image denoising module (compared to the proposed convolutional low-rank optimization module) except model size?

3. In Fig. 2, it mentions "inter-modal recovery" and "intra-modal recovery and fusion". Should these be "intra-model recovery" and "inter-model recovery and fusion"?

4. Why are the 5 image fusion metrics used in this paper selected out of tens of image fusion metrics in the literature?

5. More recent methods, i.e., those published in late 2024 and 2025 should be compared.

**Ethical Concerns:**

["NO or VERY MINOR ethics concerns only"]

**Final Justification:**

I have read the rebuttal and discussed with the authors. Thanks for the rebuttal, which is helpful for solving some of my concerns. Therefore, I changed my rating to a better level.

However, I am still not fully convinced regarding the motivation and practical usage of this method. I think the proposed method is good as a publication, but not very useful in robots, autonomous driving, and many other practical applications. The authors can try to capture some RGB and thermal images using low-cost RGB-T or thermal cameras, and then test the proposed method.

**Limitations:**

Yes, limitations are discussed in the supplementary material.

**Paper Formatting Concerns:**

I haven't found these issues.

**Quality:**

2

**Strengths And Weaknesses:**

Strengths:
- This paper is well-written and easy to follow
- The joint denosing and fusion module is interesting
- Extensive experimental results are provided
- Image qualities are good


Weaknesses:
- The motivation of this work is not very clear. As mentioned in the Experiments part, there are no existing noisy infrared and visible datasets, thus the authors needed to simulate one. Additionally, there are many existing infrared and visible image datasets that contain "clean" images, indicating the infrared and visible images captured are generally not noisy. Given these, I am not sure why noisy infrared and visible image fusion is necessary. The need of creating noisy images manually and then develop a method to fuse noisy images is not clear. This motivation should be explained more clearly. Especially, the authors should explain very clearly that fusing noisy visible and infrared images is necessary and useful in practise. Otherwise, it is likely that this paper solves a "fake" issue.

- It looks like that the proposed convolutional low-rank optimization module is a pre-processing module to generate clean data, which is then used as supervision. It is not clear why the authors do not use an unsuperivsed image denoising module. What is the disadvantage of using an unsupervised image denoising module (compared to the proposed convolutional low-rank optimization module) except model size? It seems that this kind of experiments and discussions are missing.

- The compared methods are a bit outdated. Most of the compared methods were published before 2024. More recent methods, such as those published in 2024 and 2025 should be compared.

- The selection of image fusion metrics are not justified. There are many available metrics, and a method can show different performance when different set of metrics are used. It is not clear why the authors selected the 5 metrics used in the paper. More justifications should be provided that the selected set of metrics can comprehensively evaluate image fusion methods.

---

> ### Author Rebuttal · Authors · 2025-07-30
>
> >**Q1: Explain the necessity and motivation of fusing noisy infrared and visible images.**
>
> **A1**: We appreciate this constructive suggestion. We want to declare that the clean publicly available datasets do not reflect real-world acquisition. Methods should account for inherent noise from affordable hardware.
>
> 1. Dataset quality does not reflect real-world feasibility
>
> Existing datasets are collected for the target of high image quality, thus relying on cost-prohibitive imaging systems to ensure high signal-to-noise ratios. For example, the platform of LLVIP is DS-2TD8166BJZFY-75H2F/V2, a binocular camera platform priced over $19,000. It not only has high-quality signal acquisition hardware devices, but also embeds dedicated and professional processing pipelines to enhance raw inputs, including adaptive AGC for dynamic range compression, DDE for nonlinear enhancement, and 3D DNR for denoising.
>
> However, in practice, most deployed multi-modal systems use affordable devices that may cost only several hundred dollars. The imaging systems may introduce critical distortions, such as degraded signal acquisition (low-cost microbolometers and CMOS suffer from more obvious noise), and ineffective embedded processing pipelines. These factors will result in a large amount of noise in the collected signal, which is not as high-quality as the dataset.
>
> 2. Artificial bias
>
> Existing dataset is dedicated to providing benchmarks for evaluating fusion performance. As declared, the construction of dataset involves rigorous manual screening. Some initially captured samples with low quality maybe discarded. It creates a distributional gap: real-world images contain various noise absent in curated data.
>
> We will add the discussion on necessity and motivation of fusing noisy images in the final version.
>
>
> >**Q2: Feasibility and disadvantages of replacing the convolutional low-rank optimization module with an unsupervised denoising model.**
>
> **A2**: We sincerely appreciate this insightful question, which crucially enhances the methodological rigor and expositional clarity of our work. The convolutional low-rank optimization module pre-inferences clean data as i) physics-driven prior injector to regularize fused images, and ii) implicit teachers that enable knowledge distillation of denoising priors. It seems like a pre-processing module but in fact, it is a core enabler of physics-aware and training-free supervision.
>
> It is possible to replace it with an unsupervised denoising model. However, the substitution may introduce critical disadvantages:
>
> 1. Theoretical grounding and interpretability
>
> Convolutional low-rank optimization is theoretically grounded in mathematical foundations and explicit priors, including matrix decomposition and low-rank representation theory. The objective function, constraints, and convergence of iterations are interpretable and theoretically guaranteed. Unsupervised denoising models rely on heuristic architectures with less rigorous justification for denoising mechanisms. Although certain priors can be embedded in network architecture, the success relies more on empirical design, preventing theoretical explanation and formal verification.
>
> 2. Training-data dependency
>
> Convolutional low-rank optimization is training-free, operating directly on individual noisy inputs without the need of prior dataset. It is critical for unique or rare images lacking training samples. In contrast, while unsupervised denoising models avoid paired supervision, they still demand large training data, remaining fundamentally data-driven. The performance depends on training data distribution.
>
> 3. Overfitting risks and generalization
>
> As convolutional low-rank optimization is training-free to per image, it does not rely on learning dataset-specific patterns, exhibiting robustness to out-of-distribution cases and eliminating overfitting risks. Unsupervised denoising models employ fixed networks trained on finite data distributions. When encountering noise types or content underrepresented in training, the learned mappings may fail to generalize, leading to degradation.
>
> We will add the discussion in the final version.
>
> >**Q3: Compare with more SOTA methods published in 2024 and 2025.**
>
> **A3**: Thanks for your helpful suggestion. To further strength the comparative analysis, besides comparing with Text-IF and DRMF published in 2024, we expand the competitors with three state-of-the-art methods published in 2025, including FusionBooster [r1], PromptFusion[r2] and OmniFuse [r3]. FusionBooster and PromptFusion are combined with CTNet for pre-denoising while OmniFuse is degradation-aware image fusion methods.
>
> [r1] FusionBooster: A Unified Image Fusion Boosting Paradigm. International Journal of Computer Vision, 2025.
>
> [r2] PromptFusion: Harmonized Semantic Prompt Learning for Infrared and Visible Image Fusion. IEEE/CAA Journal of Automatica Sinica, 2025.
>
> [r3] OmniFuse: Composite Degradation-Robust Image Fusion with Language-Driven Semantics. IEEE Transactions on Pattern Analysis and Machine Intelligence, 2025.
>
> As qualitative comparison is not available, we report quantitative results under four conditions (different datasets with different types of noise) below. Our method achieves optimal performance on most metrics. Less optimal performances of CT.+FusionBooster and CT.+PromptFusion stem from the focus on addressing visual degradation caused by distortions of critical features and details, prioritizing texture preservation. However, CTNet's limited denoising capability results in residual noise erroneously amplified in fusion results. The comparatively lower results of OmniFuse stem from the unified model for multiple types of degradations and supervised paradigm.
>
> |Gaussian|||LLVIP|||||M3FD|||
> |-|-|-|-|-|-|-|-|-|-|-|
> |Metrics|SSIM$\uparrow$|PSNR$\uparrow$|FSIM$\uparrow$|CC$\uparrow$|BRISQUE$\downarrow$|SSIM$\uparrow$|PSNR$\uparrow$|FSIM$\uparrow$|CC$\uparrow$|BRISQUE$\downarrow$|
> |CT.+FusionBooster|0.270|11.694|0.760|0.679|$\underline{26.381}$|0.459|10.618|0.744|$\underline{0.565}$|**16.474**|
> |CT.+PromptFusion|$\underline{0.433}$|$\underline{15.811}$|$\underline{0.803}$|$\underline{0.702}$|36.876|0.474|$\underline{15.007}$|$\underline{0.777}$|0.470|25.837|
> |OmniFuse|0.377|12.642|0.731|0.642|41.046|$\underline{0.540}$|14.488|0.763|0.459|46.787|
> |Deno-IF (ours)|**0.616**|**17.070**|**0.811**|**0.738**|**16.725**|**0.585**|**15.637**|**0.788**|**0.563**|$\underline{24.298}$|
>
> |Speckle|||LLVIP|||||M3FD|||
> |-|-|-|-|-|-|-|-|-|-|-|
> |Metrics|SSIM$\uparrow$|PSNR$\uparrow$|FSIM$\uparrow$|CC$\uparrow$|BRISQUE$\downarrow$|SSIM$\uparrow$|PSNR$\uparrow$|FSIM$\uparrow$|CC$\uparrow$|BRISQUE$\downarrow$|
> |CT.+FusionBooster|0.338|11.909|0.737|$\underline{0.683}$|$\underline{29.446}$|0.516|10.950|0.774|$\underline{0.612}$|$\underline{21.178}$|
> |CT.+PromptFusion|0.381|$\underline{15.212}$|$\underline{0.787}$|0.681|48.286|0.522|$\underline{15.881}$|$\underline{0.784}$|0.521|21.837|
> |OmniFuse|$\underline{0.479}$|12.859|0.741|0.665|39.696|$\underline{0.585}$|15.221|0.767|0.523|44.850|
> |Deno-IF (ours)|**0.633**|**16.528**|**0.805**|**0.738**|**9.280**|**0.656**|**16.525**|**0.813**|**0.648**|**16.514**|
>
> The computational complexity and efficiency of these SOTA methods and the proposed Deno-IF are reported below. Our method achieves lower computational complexity and higher inference efficiency.
>
> |Complexity|FusionBooster|PromptFusion|CT.+FusionBooster|CT.+PromptFusion|OmniFuse|Deno-IF (ours)|
> |-|-|-|-|-|-|-|
> |Para (m)|0.56|7.44|$\underline{51.97}$|58.91|98.15|**1.43**|
> |GFLOPs|241|36877|41845|78481|$\underline{5471}$|**115**|
> |Runtime (s)|2.49|1.08|15.96|14.55|$\underline{4.53}$|**0.06**|
>
> We will add the results with [r1]-[r3] in the final version to improve the previous comparison.
>
> >**Q4: Reason of selecting the five evaluation metrics.**
>
> **A4**: Thanks this question. Traditional fusion metrics can be categorized into similarity-based metrics (evaluating similarity between fused and clean source images) and statistics-based metrics (measuring fused image characteristics, such as SD, SF, MG, EN, etc.). However, when fusing noisy images, statistics-based metrics may produce misleading results as noise residuals can artificially inflate values (e.g., higher EN/SD/SF caused by noise rather than true useful information). We avoid this issue from two aspects.
>
> 1. Similarity-based metrics
>
> We prioritize similarity-based metrics that explicitly measure information fidelity over statistical properties to ensure reliable and robust evaluation. The selected metrics, i.e., PSNR, SSIM, CC and FSIM, provide complementary assessments of: i) absolute intensity preservation, ii) structural and texture similarity, iii) global feature distribution consistency, and iv) frequency-domain characteristic retention. The orthogonal combination of these multi-dimensional metrics offers a comprehensive evaluation of information fidelity.
>
> 2. No-reference quality-based metrics
>
> To assess image quality, we employ BRISQUE instead of conventional statistics-based metrics. By analyzing natural scene statistics through normalized pixel intensity distributions, BRISQUE detects unnatural distortions via mean-subtracted contrast-normalized coefficients and generalized Gaussian modeling. Its trained regression model penalizes noise-induced deviations from natural image statistics, yielding scores consistent with human perception and reliable for noisy scenarios.
>
> >**Q5: Correct “inter-modal recovery” and “intra-modal recovery and fusion” in Fig. 2.**
>
> **A5**: Thanks for careful reading and catching the terminology inconsistency. The correct terms should be "intra-modal recovery" and "inter-modal recovery and fusion". The terms “inter-” and “intra-” are mistakenly swapped while “modal” is correct as it refers to distinct imaging modalities rather than “model”. We will carefully correct all instances in the final version.

---

> ### Comment · Reviewer_GvvZ · 2025-08-06
>
> Thank you for the rebuttal, which addressed some of my concerns.
>
> However, I am still not convinced by the authors regarding the motivation.
>
> It is well-known that thermal cameras are expensive. Thermal cameras at hundreds of dollars are basically not very useful in practise. Do you mean that, by using your method, people can use devices of several hundreds of dollars to achieve the performance of devices of thousands of dollars? To be honest, I doubt this point.
>
> Additionally, thermal camera is usually applied in applications where pedestrian safety is a critical issue. In these applications, such as autonomous driving, this is not the case - "most deployed multi-modal systems use affordable devices that may cost only several hundred dollars". In these applications, people tend to use high-quality camera to capture better images to ensure safety. According to my data capturing experience, I don't think fusion with noisy images is necessary in these cases when high-quality thermal cameras are used in these important applications. Therefore, I still think this paper creates a "fake" issue and then propose a solution to this "fake" issue. I think this is the main problem of this paper. I am not sure this method is useful in practise, such as in autonomous driving.
>
> Furthermore, would it be possible to provide fusion results in terms of other fusion metrics other than the five utilized in the paper? This will show more comprehensive comparison.

---

> > ### Author Response · Authors · 2025-08-07
> >
> > We appreciate your valuable comments, especially for your insightful thoughts regarding the noisy dataset and problem authenticity.
> >
> > **First**, we fully agree that high-quality devices possess irreplaceable hardware advantages in absolute performances. Our goal is not to claim full equivalence between low-cost and premium devices, which would indeed be unrealistic. Rather, the core value lies in enhancing information extraction, recovery, and fusion from noisy, low-cost infrared and visible sensors for specific tasks in challenging environments via advanced fusion methods.
> >
> > **Additionally**, we fully acknowledge that there is a tendency to use high-quality cameras in safety-critical domains such as autonomous driving. However, we would like to further elaborate on the motivation and real-world significance of infrared and visible image fusion under noisy conditions:
> > 1. **Ubiquity and Inevitability of Noise in Real-World Scenarios**: Even high-quality cameras will inevitably encounter challenging environmental conditions (e.g., extreme lighting, extreme temperatures, etc.) that push the performance limits. In such cases, even high-quality cameras may produce images containing noise. Therefore, it is unrealistic to always assume that input images are perfect.
> > 2. **Widespread Cost-Limited Application Scenarios**: While autonomous driving represents a typical application, image fusion has an extremely broad range of applications. Many scenarios are cost-constrained and cannot deploy high-quality imaging equipment (e.g., widely deployed civilian and industrial surveillance cameras typically use lower-cost sensors), yet they still require reliable perception.
> > 3. **Core Research Value of Robustness Enhancement**: The primary goal of this study is to explore and improve the robustness of fusion methods when facing noise interference. Investigating model robustness through simulated degradation is a widely accepted and applied research paradigm, and we follow this paradigm to design our work.
> >
> > **Finally**, we provide the results evaluated with other fusion metrics as below. As no-reference quality-based metrics are prone to degradation and may become artificially high due to residual degradation. Therefore, we use similarity-based metrics to measure information fidelity. The additional fusion metrics include:
> > 1. $N^{AB/F}$ defines the normalized absolute deviation factor to measure the artifacts introduced during the fusion process [r1].
> > 2. $Q^{AB/F}$ measures the edge information transferred from source images to the fused image [r2].
> > 3. VIF quantifies the perceptual quality of fused images via visual information fidelity. It is based on human vision system and measures how faithfully the critical information from source images is visually preserved [r3].
> >
> > |LLVIP| CT.+Swin. | CT.+Tar. | CT.+CDD. | CT.+DDFM| CT.+Meta. | DRMF | Text-IF | CT.+Fusion. | CT.+Prompt. | OmniFuse | Deno-IF |
> > |--|--|--|--|--|--|--|--|--|--|--|--|
> > | $N^{AB/F}\downarrow$ | 0.091 | 0.062 | 0.083 | $\underline{0.050}$ | 0.098 | 0.239 | 0.176 | 0.203 | 0.087 | 0.077 | **0.014** |
> > |$Q^{AB/F}\uparrow$|0.296| 0.187 | 0.284 | 0.194 | 0.229 | 0.262 | **0.331** | 0.240 | 0.285 | 0.251 | $\underline{0.302}$ |
> > |VIF$\uparrow$| 0.136 | 0.114 | 0.134 | 0.133 | 0.131 | 0.135 | $\underline{0.163}$ | 0.124 | 0.136 | 0.148 | **0.170** |
> >
> > |M3FD| CT.+Swin. | CT.+Tar. | CT.+CDD. | CT.+DDFM| CT.+Meta. | DRMF | Text-IF | CT.+Fusion. | CT.+Prompt. | OmniFuse | Deno-IF |
> > |--|--|--|--|--|--|--|--|--|--|--|--|
> > | $N^{AB/F}\downarrow$ | 0.053 | 0.074 | 0.077 | 0.052 | 0.129 | 0.140 | 0.172 | 0.099 | 0.085 | $\underline{0.044}$ | **0.008** |
> > |$Q^{AB/F}\uparrow$| 0.285 | 0.194 | 0.271 | 0.194 | 0.266 | **0.446** | $\underline{0.401}$ | 0.233 | 0.276 | 0.316 | 0.320 |
> > |VIF$\uparrow$| 0.127 | 0.114 | 0.127 | 0.126 | 0.132 | $\underline{0.163}$ | 0.162 | 0.118 | 0.129 | 0.150 | **0.163** |
> >
> > As measured on LLVIP and M3FD datasets under Gaussian noise, our method achieves the highest $N^{AB/F}$ and VIF values on both datasets, demonstrating exceptional robustness in preserving source information fidelity and visual naturalness against noise interference. While attaining competitive second/third places in $Q^{AB/F}$, it stems from our emphasis on suppressing high-frequency noise components during edge transfer. It is a trade-off to prioritize perceptual coherence over maximizing absolute edge metrics.
> >
> > [r1] Objective Image Fusion Performance Characterisation. Proceedings of the IEEE International Conference on Computer Vision, 2005.
> >
> > [r2] Objective Image Fusion Performance Measure. Electronics Letters, 2000.
> >
> > [r3] A New Image Fusion Performance Metric Based on Visual Information Fidelity. Information Fusion, 2013.

---

> > > ### Comment · Reviewer_GvvZ · 2025-08-08
> > >
> > > Thank you for the rebuttal.

---

### Official Review · Reviewer_8daB · 2025-07-06

**Clarity:** 3
**Significance:** 3
**Originality:** 3
**Rating:** 5
**Confidence:** 4

**Summary:**

The paper proposes a two-stage framework for visible-infrared image fusion under noisy conditions. It integrates a convolutional low-rank optimization module for intra-modal denoising and a joint fusion network for cross-modal feature integration. By leveraging spatial alignment and analogical reasoning, the method enhances feature consistency and achieves superior performance across various noise levels.

**Questions:**

1.	In Figure 2, it is unclear whether the operator “c” represents feature concatenation, element-wise multiplication, or addition. This should be explicitly annotated in the figure.
2.	In the fusion stage, it appears that the visible and infrared images are simply overlaid. How is noise suppression handled in this process?
3.      In Subsection 3.3, the phrase “inter-modal recovery and intra-modal recovery and fusion” exhibits structural ambiguity, which may lead to misinterpretation. It is recommended that the authors revise the wording throughout the manuscript to enhance clarity and ensure the precise use of terminology.

**Ethical Concerns:**

["NO or VERY MINOR ethics concerns only"]

**Limitations:**

Yes

**Paper Formatting Concerns:**

It is recommended that the term "convolutional neural network (CNN)" be capitalized when used with its abbreviation to maintain consistency and adhere to standard academic conventions.

**Quality:**

3

**Strengths And Weaknesses:**

Strengths
The proposed unsupervised image fusion framework introduces a novel perspective by jointly addressing denoising and fusion in noisy visible-infrared scenarios, which is a less explored yet practically important problem.
The integration of a convolutional low-rank optimization module enables effective extraction of clean components from noisy inputs without requiring paired training data, improving generalization across varying noise conditions.
The experimental section is comprehensive, covering diverse and variable noise settings, and convincingly demonstrates the effectiveness and robustness of the proposed method.
Weaknesses
Some aspects of the method, particularly the implementation and interplay between the convolutional low-rank optimization and the joint fusion network, are not clearly described and may hinder reproducibility.
The proposed method adopts a two-stage architecture, which, while modular in design, may lead to a more prolonged training process and reduced overall training efficiency. It is recommended to optimize the training strategy or provide an analysis of computational cost to enhance practical applicability.

---

> ### Author Rebuttal · Authors · 2025-07-30
>
> >**Q1:  Implementation and interplay between the convolutional low-rank optimization and joint fusion network.**
>
> **A1**: We sincerely appreciate this constructive suggestion. We want to emphasize that the connection between these modules operates fundamentally via deep implicit physics-driven constraint propagation. The ultimate goal is to obtain a joint denoising and fusion module via an unsupervised approach, yet struggles to infer high-quality structural cues to guide network optimization.
>
> To address it, the convolutional low-rank optimization module leverages convolutional low-rank priors to derive clean components. These components serve as i) physics-driven prior injector to regularize fused images, and ii) implicit teachers that enable knowledge distillation of denoising priors directly into the final joint denoising and fusion network. The design enables simultaneous noise suppression and information fusion while eliminating the need for low-rank module reuse during testing, thereby enhancing real-time inference efficiency.
>
> We will emphasize the deep implicit connection and include a detailed algorithmic presentation in pseudocode format in the final version to enhance methodological clarity.
>
> >**Q2: Optimize the training strategy or provide an analysis of computational training cost.**
>
> **A2**: Thanks for your insightful suggestions. We first perform an analysis of computational cost. As the training latency is dominated by the first stage (convolutional low-rank optimization module), we quantify the computational bottleneck within this module. In this module, computational complexity scales primarily with i) convolution kernel size ($k_1,k_2$) and ii) number of iterations ($T$) for solving the low-rank subproblem. The training costs of different settings are reported below. We observe that increasing $k_1,k_2$ incurs significant increase in training time while the number of iteration scales linearly with computational cost. Nevertheless, the entire training process remains practical, completing within several days.
>
> |$k_1,k_2$|12|18|24|30|36|42|
> |--|--|--|--|--|--|--|
> |Training time/epoch (m)|41.31|54.35|71.51|93.28|121.22|151.68|
>
> When $k_1,k_2=12$, we change the number of iterations (T) and quantify the training cost.
> |$T$|10|20|30|40|50|60|
> |--|--|--|--|--|--|--|
> |Training time/epoch (m)|20.58|30.85|41.31|51.61|61.62|71.62|
>
> A possible solution to optimize the training strategy is to adaptively adjust $k_1$ and $k_2$ based on the scene content, and dynamically update $T$ according to the convergence behavior. We will report the training cost in the final version. In future work, we plan to further develop adaptive strategies for determining convolutional kernels and iterative updates to improve training efficiency.
>
> >**Q3: Annotation of operator “c” in Figure 2.**
>
> **A3**: Thanks for this question. The operator “c” represents feature concatenation. We will add the explicit annotation in Figure in the final version.
>
> >**Q4: How noise suppression is handled in the fusion stage.**
>
> **A4**: We appreciate this opportunity to clarify our fusion methodology. We want to clarify that the fusion process is not a simple overlay of source images, but rather a sophisticated extraction, recovery, fusion, and reconstruction process implemented through the joint denoising and fusion module.
>
> From the aspect of network architecture, noise is suppressed through Transformer blocks in both intra-modal recovery and inter-modal recovery and fusion.
>
> From the aspect of network optimization, clean components derived with convolutional low-rank priors serve as i) physics-driven prior injector to regularize fused images, and ii) implicit teachers that enable knowledge distillation of denoising priors into the fusion network. It enables simultaneous noise suppression and information fusion. In the final version, we will emphasize the implementation of denoising functionality in the network more prominently to prevent potential misunderstanding.
>
> >**Q5: Revise the phrase “inter-modal recovery and intra-modal recovery and fusion”.**
>
> **A5**: Thanks for your careful reading and reminder. The terms “inter-” and “intra-” are mistakenly swapped in the original manuscript. The correct terms should be "intra-modal recovery" and "inter-modal recovery and fusion". We will carefully revise the wording through the manuscript and ensure the precise use of terminology in the final version.

---

> > ### Comment · Reviewer_8daB · 2025-08-05
> >
> > I have read the authors' rebuttal. The authors have addressed my concerns clearly and thoroughly. I am satisfied with their response and will keep my original score.

---

### Decision · Program_Chairs · 2025-09-17

**Decision:**

Accept (spotlight)

**Comment:**

This paper presents Deno-IF, an unsupervised method for fusing noisy visible and infrared images. The core idea is to jointly perform denoising and fusion. This is achieved through two main components: a convolutional low-rank optimization module that extracts clean image components from noisy inputs, and a unified network that then performs the fusion.

The strengths of the paper lie in its novel unsupervised approach to a practical problem, addressing noise in real-world fusion scenarios without needing paired clean data. The framework is technically sound, and the use of a convolutional low-rank prior as a loss function is an interesting contribution. The experimental validation is thorough and demonstrates the method's effectiveness.

Initial reviews raised concerns about the motivation (questioning the prevalence of noisy real-world data), the clarity of certain methodological details, and the selection of comparison methods. The authors provided a comprehensive rebuttal, successfully clarifying the real-world necessity of handling noise from affordable sensors, explaining the interplay between modules, and adding comparisons with more recent state-of-the-art methods. These responses satisfied the reviewers, leading to a consensus for acceptance.